# Digital proximity tracing on empirical contact networks for pandemic control

G. Cencetti[1,10], G. Santin [1,10], A. Longa [1,2], E. Pigani [1,3], A. Barrat [4,5], C. Cattuto[6,7], S. Lehmann [8], M. Salathé[9] & B. Lepri [1✉]

Digital contact tracing is a relevant tool to control infectious disease outbreaks, including the COVID-19 epidemic. Early work evaluating digital contact tracing omitted important features and heterogeneities of real-world contact patterns influencing contagion dynamics. We fill this gap with a modeling framework informed by empirical high-resolution contact data to analyze the impact of digital contact tracing in the COVID-19 pandemic. We investigate how well contact tracing apps, coupled with the quarantine of identified contacts, can mitigate the spread in real environments. We find that restrictive policies are more effective in containing the epidemic but come at the cost of unnecessary large-scale quarantines. Policy evaluation through their efficiency and cost results in optimized solutions which only consider contacts longer than 15–20 minutes and closer than 2–3 meters to be at risk. Our results show that isolation and tracing can help control re-emerging outbreaks when some conditions are met: (i) a reduction of the reproductive number through masks and physical distance; (ii) a low-delay isolation of infected individuals; (iii) a high compliance. Finally, we observe the inefficacy of a less privacy-preserving tracing involving second order contacts. Our results may inform digital contact tracing efforts currently being implemented across several countries worldwide.

[1] Fondazione Bruno Kessler, Trento, Italy. [2] University of Trento, Trento, Italy. [3] University of Padua, Padua, Italy. [4] Aix Marseille Univ, Université de Toulon, CNRS, CPT, Turing Center for Living Systems, Marseille, France. [5] Tokyo Tech World Research Hub Initiative (WRHI), Tokyo Institute of Technology, Tokyo, Japan. [6] University of Turin, Turin, Italy. [7] ISI Foundation, Turin, Italy. [8] Technical University of Denmark, Copenhagen, Denmark. [9] École Polytechnique Fédérale de Lausanne (EPFL), Lausanne, Switzerland. [10] These authors contributed equally: G. Cencetti, G. Santin. ✉email: lepri@fbk.eu

As of mid-January 2021, the COVID-19 pandemic has resulted in over 85 million detected cases worldwide[1], overwhelming the healthcare capacities of many countries and thus presenting extraordinary challenges for governments and societies[2,3]. Rigorous restrictions such as lockdowns and quarantine have proven to be effective in many countries as a measure to curb the spread of COVID-19, limit contagions and reduce the effective reproductive number[2,4–12]. Many areas slowly started to lift the restrictions, but new outbreaks appeared again, arriving in waves as anticipated by several early models[13,14]. An effective and affordable long-term plan is required, since the fraction of the population that has been infected is still far too low to provide herd immunity[3].

Despite their efficacy, large-scale quarantine and lockdown strategies carry large costs[5]. Moreover, in a situation where most of the population is not infected, population-wide lockdowns are far from optimal, and interventions at smaller scale, selectively targeting individuals at higher risk of spreading the disease, are more desirable.

While the testing and isolation of symptomatic cases is crucial, it is insufficient in the case of COVID-19, since there is clear evidence of presymptomatic and asymptomatic transmission[15–19]. Thus, the identification and isolation of infected cases must be coupled with a strategy for tracing their contacts and preventively quarantining them[17,20–22]. Traditional manual contact tracing, besides being slow and labor intensive[23–25], is not able to entirely reconstruct close proximity contacts[26,27]. Thus, technologies based on digital sensors have been developed to complement manual tracing. The idea is to leverage the widespread dissemination of smartphones to develop proximity-sensing apps based on the exchange of Bluetooth radio packets between them[17,28–33], within a privacy-preserving contact tracing framework[28].

The efficacy of digital contact tracing (DCT)[20–22,34–41] has been discussed in several recent papers. We draw inspiration from the work by Fraser et al.[42], recently adapted to the case of COVID-19 by Ferretti et al.[17]. These work models the pandemic evolution using recursive equations describing the number of infected individuals in a homogeneously mixed population, taking into account the evolving infectiousness of the infected individuals. The analysis is based on two effective parameters, $\varepsilon_I$ and $\varepsilon_T$, to represent the ability to identify and isolate infected individuals, and to correctly trace their contacts, respectively. Assuming an exponential growth for the number of infected individuals (applicable in early phases of an uncontrolled epidemic outbreak) the authors studied how the growth rate depends on these intervention parameters.

Here, to better understand the effectiveness of real-world contact tracing, we expand this approach.

First, we restructure and generalize the mathematical framework to allow us to completely avoid assumptions regarding the functional form of the epidemic growth. This development makes the setting applicable to any possible evolution shape and any phase of the epidemic. Moreover, we modify the epidemiological aspects of the model according to the recent literature on COVID-19[43–45], to properly consider asymptomatic cases and the delay in isolating individuals after they are identified as infected. We consider different values of $R_0$, reduced with respect to the one assigned to the free pandemic, to take into account the widely implemented additional containment strategies, e.g., physical distancing and wearing masks (Supplementary Note 2).

Second, we provide a realistic quantification of the tracing ability $\varepsilon_T$ by performing simulations of contact tracing strategies on real-world data sets collected across different social settings (i.e., a university campus, a workplace, a high school)[46–48]. Hence, the tracing ability $\varepsilon_T$, defined by Ferretti et al.[17] as a free parameter, becomes here an empirically estimated quantity,

which directly depends on the contact network. The impact of the tracing procedure on the spread can then be evaluated by inserting $\varepsilon_T$ into the mathematical model.

Third, we assume that the probability of a contagion event occurring during an interaction between a susceptible and an infected individual also depends on the duration and on the degree of proximity of the contact[49,50] (along with other epidemiological variables such as the infectiousness of the individual). This can be simulated on real contact data sets, in particular on the Copenhagen Networks Study (CNS) data set[46] that provides proximity information, via the strength of Bluetooth radio packets exchanged between their smartphones.

Finally, we investigate in detail the contact tracing procedure, designing appropriate policies in terms of the definition of the most risky contacts. We thus implement a system where tracing does not necessarily imply a massive preventive quarantine of the population. We define duration and proximity thresholds to discriminate between "risky" contacts and contacts that instead correspond to a low contagion probability. Note that, as contagion events are stochastic in nature, not all contacts that we consider at risk lead to infection events. This leads to "false positives", i.e., non-infected individuals who will be quarantined. Similarly, among the contacts considered as "non-risky" by the contact tracing, some might actually have led to a contagion event ("false negatives"). Quantifying these outcomes represents crucial information to calibrate the policies for contact tracing apps. Quarantine too few and omit many potential spreaders. Quarantine too many and incur unnecessarily high social costs.

Overall, our approach allows to evaluate the effect of different contact tracing policies, not only on the disease spread but also in terms of their impact on the population, as quantified by the fraction of quarantined individuals.

## Results

### A modeling framework for DCT on empirical contact networks. In this section, we introduce our model for contact tracing. The tracing procedure allows to identify individuals who are considered to be at the highest infection risk, and to quarantine them without necessarily isolating a large fraction of the population. This allows devising ad hoc strategies to control the epidemic.

We consider a population within which a virus is spreading, and the spread is determined by the contacts between individuals. As we do not consider geography nor large-scale mobility, our modeling can be considered as referring to a limited geographical area or community, similar to previous modeling efforts[17,21]. The spreading process is designed in order to mimic the COVID-19 epidemic, thus characterized by values of $R_0$, viral load and fraction of asymptomatic individuals that are typical of SARS-CoV-2. We assume that two types of non-pharmaceutical interventions are at play: isolation and contact tracing. Infected individuals are isolated when they self-report as symptomatic or if they are identified through randomized testing. Isolated individuals do not have contact with other individuals, thus can not infect anyone else once they have been identified. In other words, they are removed from the system. Individuals who have had potentially contagious contact with identified infected individuals are traced and can be warned through a privacy-preserving app on their smartphone[28], and they quarantine preemptively.

The only difference between isolation and quarantine is that the latter is only precautionary: if quarantined individuals show symptoms before the end of quarantine they immediately become isolated and their past contacts (before quarantine) are traced, otherwise they are released at the end of the quarantine.

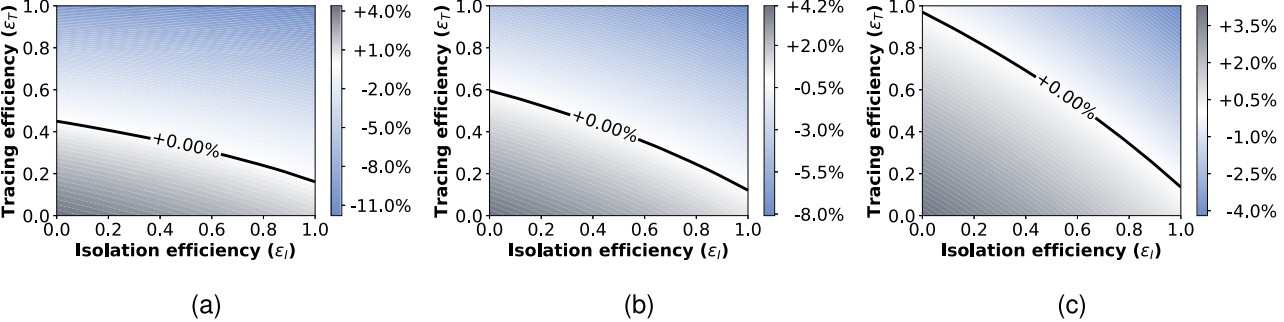

**Fig. 1 Infection rate scenarios.** Growth or decrease rate of the number of newly infected individuals, assuming either that all the infected people can eventually be identified and isolated (**a**); or that only symptomatic people can be isolated with 20% of asymptomatic infected individuals (**b**); or that only symptomatic people can be isolated with 40% of asymptomatic infected individuals (**c**). Infection rates are reported as a function of the isolation efficiency $\varepsilon_I$ and the tracing efficiency $\varepsilon_T$. In all the three settings the cases are reported with a delay of 2 days.

A natural baseline for the work we present here is the model by Fraser et al.[42], recently adapted to the COVID-19 case in Ferretti et al.[17]. The mathematical model is based on recursive equations designed to quantify the number of newly infected individuals at time intervals, given a characterization of the disease in terms of infectiousness and manifestation of symptoms. The model is designed to consider the two interventions described above, whose effectiveness are quantified by two parameters $\varepsilon_I, \varepsilon_T$ varying from 0 to 1, where $\varepsilon_I = 0$ means "no isolation" and $\varepsilon_I = 1$ represents a perfectly successful identification and isolation of all infected individuals; analogously, $\varepsilon_T$ quantifies the efficacy of contact tracing.

Here we use this model as a stepping stone in order to define a more general approach. The generalization of the equations of Fraser et al.[42] is derived in detail in the Supplementary Information and resolves an important limitation. Indeed, it identifies a solution at finite time $t$, while the original model only shows the asymptotic behavior, for $t$ going to infinity. The equation models the number $\Lambda(t, \tau)$ of people who are infected at time $t$ by people that have been in turn infected for a time $\tau \leq t$. In the equation, $R_0$ is the reproductive number of the disease, $\omega(\tau)$ is the infectiousness of individuals at time $\tau$ after being infected, and $s(\tau)$ is the probability of symptom onset at time $\tau$ after infection. The details of each of these quantities are discussed in Supplementary Note 1.1. The equation reads

$$\Lambda(t, \tau) = R_0 \omega(\tau)(1 - \varepsilon_I s(\tau)) \int_0^{t-\tau} \left( 1 - \varepsilon_T \frac{s(\rho + \tau) - s(\rho)}{1 - s(\rho)} \right) \Lambda(t - \tau, \rho) d\rho,$$
(1)

where the integration variable $\rho$ spans the time range between 0 and $t - \tau$, meaning that the contagion at time $t$ from people infected at time $t - \tau$ is in turn affected by contagion at time $\rho$ before $t - \tau$.

For $\varepsilon_I = 0$ and $\varepsilon_T = 0$ we obtain a free spreading without control. The quantity of interest, which can be derived by numerically solving the above equations, is the incidence $\lambda(t) := \int_0^t \Lambda(t, \tau) d\tau$ of newly infected individuals at time $t$. We use the model to predict the evolution of $\lambda(t)$ up to time $t = 50$ days, which is sufficient for the numerical solutions to reach a stationary growth or decline regime (constant growth or decline rate of $\lambda(t)$), and we consider the average growth or decline in the last 10 days as an indicator of the long-term behavior of the epidemic. A negative number indicates that the epidemic is declining, while a positive one corresponds to growth (uncontained epidemic).

An important feature of the model is given by the probability $s(\tau)$. The ideal case in which all infected individuals can

eventually be identified because they exhibit symptoms ($s(\tau)$ approaching 1 for large times) is reported in Fig. 1a: this represents the best-case scenario, considered in the previous studies of this model[17,42]. Next, we assume instead that 40% of infected individuals are asymptomatic[17–19,51,52] and that only symptomatic individuals can be identified: no randomized testing is performed. This represents our worst-case scenario. We represent the presence of asymptomatic individuals by considering that the probability of an infected individual to display symptoms is a growing function of time, which however never reaches 1. In this case, the model predicts epidemic containment for the upper half of the range of values of the parameters $\varepsilon_I$ and $\varepsilon_T$ (Fig. 1c).

In the following, we assume an alternative scenario where 50% of the asymptomatic individuals are identified by a policy of randomized testing[11]. These, added to the symptomatic individuals, result in a detection of 80% of the total infected cases. We remark that this scenario is equivalent to assuming that asymptomatic individuals account for only 20% of the infected population[53,54]. Indeed, there is still no agreement in the scientific community about the fraction of asymptomatic infections for COVID-19, and different possible scenarios should be considered[11] (Supplementary Note 1). This is our baseline for the following investigations and the resulting model predictions are plotted in Fig. 1b.

Note also that we take into account in all settings a delay of 2 days between the detection of an infected individual and the time when this person is actually isolated and contact tracing is implemented. A delay of 3 days is considered in Supplementary Note 3.2.

*Tracing efficiency based on empirical contact data.* The proposed mathematical framework makes it possible to address our main goal: characterizing the efficiency of contact tracing. This can be quantified by $\varepsilon_T$, which instead of being a free parameter can be estimated numerically, by observing how well the implemented policies enable to find the infected individuals. More precisely, we assume that a fraction $\varepsilon_I$ of infected individuals is identified at each time step. Their recent contacts are then traced and, according to the nature of their interaction, as we explain in detail in the next sections, some of them will be classified as "at risk and thus possibly contagious". Tracing is therefore strongly dependent on the ability to identify those primary infected individuals that caused the secondary infections, and we thus assume that $\varepsilon_T$ is proportional to $\varepsilon_I$. Moreover, it is influenced by the actual ability to find the secondary cases, given the primary infected. This in turn depends on multiple factors, involving the spreading model, the definition of a risky contact, the app adoption, the

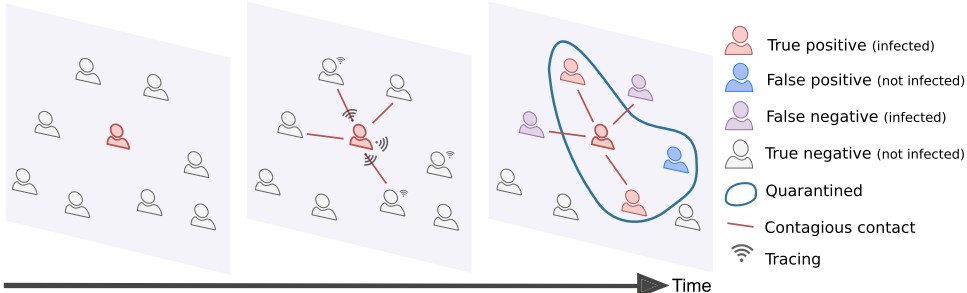

**Fig. 2 Contagion, tracing, and quarantines.** The contacts among users of the contact tracing app are registered via the app. When individuals are identified as infected they are isolated, and the tracing and quarantine policy is implemented. Depending on the policy design, the number of false positives and false negatives may vary significantly.

compliance to quarantine and clearly the quantity and nature of contacts in the population. For this reason, we need a numerical model that takes into account all these factors and simulates the spreading, with isolation and tracing, in a population of individuals with realistic contacts. To this end, we make use of three different data sets of empirical contacts involving large groups of people, in a high school, in a university campus and in an office building. The variable $\varepsilon_T$ will be computed by counting, for each primary infection, the fraction of the corresponding secondary cases that are actually quarantined according to some contact tracing strategy, see Section "Aggregation and parameter estimation" for the details on the derivation of $\varepsilon_T$.

The data that we use have been collected using wearable devices in different populations of individuals and contain time-resolved information on their pairwise close-range proximity interactions. In each case, we simulate an epidemic spread starting from a single random individual. The epidemic propagates from person to person via their interactions and we assume that the recent contacts of each individual are stored in their mobile phones. Each infected individual has a probability of being identified equal to $\varepsilon_I$. When this happens, all the identified people are isolated, i.e., removed from the simulation, and their recent stored contacts are automatically traced (i.e., warned by the app). In order to avoid quarantining a large portion of the population, we define specific criteria to determine which contacts are at risk, and only the corresponding individuals go into quarantine. As the definition of risky contacts is made a priori, and as infection events occur stochastically, quarantines will not only concern individuals who have been infected, but also some who have been in contact but were not infected (false positives), while some other individuals who have been infected although their contact were not considered at risk, will not receive any warning by the app and thus remain outside quarantines (false negatives). Note also that individuals who did not adopt the app cannot be notified nor quarantined, and contribute either to the true or to the false negatives. This is schematically explained in Fig. 2. Different policies to define the risky contacts will be delineated in Section "Design of appropriate policies" and their efficiency will be quantified by not only observing their ability in controlling the epidemic but also by their efficiency in minimizing the number of false positives, i.e., unnecessary quarantines.

In the following we will mainly rely, for the numerical evaluations of tracing, on the CNS data set[46]. These data describe the interactions of 706 students, as registered by the exchange of Bluetooth radio packets between smartphones, for a period of one month. From the complete data set we extract the proximity measures in the form of Bluetooth signal strength. We therefore have access to two important properties of contacts: their duration and the proximity of the two individuals at the time

of the interaction. We are hence able to refine the spreading model by including the dependence on these variables too, as explained in the next section. Moreover, the risk assessment in the tracing procedure will be based on contact proximity and duration thresholds, corresponding to different policies which will be discussed in Section "Design of appropriate policies".

In the Supplementary Information we also show simulations performed using two other data sets collected by the Socio-Patterns collaboration in two environments: a high school[48] and an office building[47].

It is important to emphasize that these simulations are specifically used to evaluate the impact of isolation and tracing in different contexts and under different policies and to extract the resulting values of isolation and tracing efficiencies. On the other hand, the epidemic model we use to understand which policies are efficient is the theoretical one described by Eq. (1) and is thus not restricted to any specific setting.

*How infectiousness depends on duration and proximity.* In the theoretical model (1), infectiousness is simply given by the curve $\omega(\tau)$ multiplied by $R_0$; on the other hand, as stated above, the numerical simulations make it possible to take into account several crucial factors, like duration and proximity of contacts.

We thus multiply $\omega(\tau)$ by two independent factors, $\omega_{\text{exposure}}(e)$ and $\omega_{\text{dist}}(s_s)$. They represent the probability for an infected individual to transmit the disease respectively given the duration $e$ of contact and given the signal strength $s_s$ of a contact. Here, the Bluetooth received signal strength can be considered as a proxy for the distance between two individuals, where signal attenuations (in dBm) with smaller absolute value tend to correspond to smaller distances[55]. We refer to Supplementary Note 1.2 for a detailed discussion on the functional shapes of $\omega_{\text{exposure}}(e)$ and $\omega_{\text{dist}}(s_s)$. In particular, as both are parametric functions, it is possible to tune their parameters by imposing some physical constraints regarding duration, distance, and $R_0$. The reproductive number of COVID-19 can be extracted from the literature as being close to $R_0 = 3$[45], while there is little evidence for the dependence on proximity and duration; we thus consider multiple possible infection curves corresponding to different combinations of $\omega_{\text{exposure}}(e)$ and $\omega_{\text{dist}}(s_s)$, keeping $R_0 = 3$ fixed. To this aim, we elaborate a procedure aimed at choosing the function parameters starting from physical constraints so as to always consider meaningful infectiousness curves. The procedure is explained in details in Supplementary Note 1.2, where we characterize three different possible curves. The constraint given by $R_0$ requires to find a good balance between the two functions $\omega_{\text{exposure}}(e)$ and $\omega_{\text{dist}}(s_s)$. If for instance we suppose that infectiousness is high even at long distances we should thus set $\omega_{\text{exposure}}$ such that contacts are contagious only for long durations

| | Signal strength threshold $T_p$ (dBm) | Duration threshold $T_d$ (min) | Fraction of CNS contacts |
|---|---|---|---|
| ● Policy 1 | −73 | 30 | 2.2% |
| ● Policy 2 | −80 | 20 | 7.3% |
| ● Policy 3 | −83 | 15 | 13.4% |
| ● Policy 4 | −87 | 10 | 25.9% |
| ● Policy 5 | −91 | 5 | 56.7% |

(a)

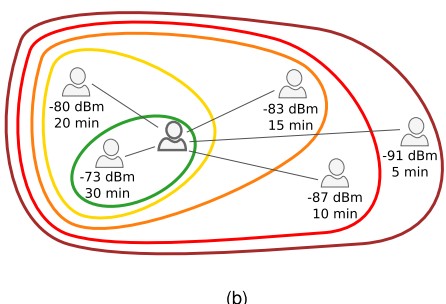

(b)

**Fig. 3 Policies based on distance and duration. (a)**: The signal strength threshold $T_p$ and the duration threshold $T_d$ defining the policies are reported. Contacts with a duration larger than $T_d$ and signal strength larger than $T_p$ are considered at risk. The last column gives the fraction of the total number of interactions of the CNS data set that they correspond to. A larger value of the magnitude of the signal strength tends to correspond to a larger distance, such that in the second column the thresholds go from the least to the most restrictive policy. The policies are sketched in (**b**).

in order not to have a huge $R_0$ (e.g., the pink curves in Supplementary Fig. 1). Vice versa, if $\omega_{\text{dist}}$ is adjusted such that only close proximity contacts are contagious, we should give more importance to duration and suppose that also short durations are at risk (e.g., the blue curves in Supplementary Fig. 1). In Supplementary Note 1.2, we show the results of simulations in these different cases. We observe that for the controllability of the epidemics, the different types of infectiousness do not lead to significant differences. However, from the point of view of cost versus the effectiveness of the restrictive measures, different curves lead to different results. We discuss this point in Supplementary Note 2.1. Here, we choose for definiteness one of the obtained pairs of curves ($\omega_{\text{exposure}}(e)$, $\omega_{\text{dist}}(s_s)$) compatible with $R_0 = 3$, and we assume in the following that infectiousness is governed by these. They correspond to an $\omega_{\text{exposure}}(e)$ which reaches 90% infectiousness after 2 h of contact, and to an $\omega_{\text{dist}}$ such that the contagion probability drops by 50% at a distance of 2.5 m, and by 99% at 7.0 meters.

Finally, in the numerical model, we rescale the curves of infectiousness of a factor $r_{R_0}$, which plays a pivotal role. Indeed, the procedure described above for parameter setting is aimed at reconstructing a scenario without restrictions, where the epidemic of COVID-19 is free to spread and is characterized by a reproductive number equal to 3. However, in this work we analyze the effect of isolation and tracing in the context of reemerging epidemics where a number of protective measures are in places, such as face masks and physical distancing. Such measures contribute to mitigate the spreading and enter in our model as an overall reduction of $R_0$, in a range suggested by recent literature[56–59]. This can be obtained by setting the reduction factor $r_{R_0}$ to specific values, reported in Supplementary Table 5 in the Supplementary Information.

*Design of appropriate policies.* As mentioned above, the empirical CNS data set provides us with the opportunity to devise policies for tracing in order to avoid a massive preventive quarantine of the population.

We can classify contacts at a low and high probability of contagion on the basis of thresholds of duration and proximity: only contacts with duration above a threshold $T_d$ and Bluetooth signal strength above a threshold $T_p$ are considered as at risk and thus stored in the individual's devices (when both individuals in contact have adopted the app). Assuming that the dependence of infectiousness from duration and proximity is unknown, we consider several possible values for the thresholds $T_d$ and $T_p$, thus defining multiple possible policies, reported in Fig. 3, from the least to the most restrictive. We also consider two additional

policies in Supplementary Note 3.5, corresponding to either close range but short exposure interactions or long-range but long exposure interactions.

We remark that the policies implement distance detection directly as a measure of the received signal strength indicator (RSSI) values, since a precise and reliable conversion to an actual distance is a notoriously difficult task[55,60] that would only add a layer of uncertainty to our analysis, without any gain in terms of accuracy. It is in general true that weak signal strengths correspond to large distances between users and vice versa but the link between RSSI and actual distance is affected by multiple factors, from the smartphone brand to the presence of obstacles between devices, and more[55,60].

In substance, we simulate the epidemic and at the same time implement the contact tracing, supposing that we do not know which individuals are infected. We then compare the set of quarantined individuals with the set of people who have actually been infected in the spreading simulation and measure the performances of each tracing policy (i.e., of each definition of thresholds $T_p$ and $T_d$). The performance of a policy is quantified first of all by its ability to find the infected individuals, and consequently by its ability to contain the epidemic according to our mathematical model; in addition, we will measure the efficacy of a policy in quarantining only infected individuals (i.e., in limiting the number of false positives), in order to limit the social and economic damage to society.

Figure 4 shows the distributions of RSSI and contact durations of the interactions contained in the CNS data set. Most contacts have a short duration and low signal strength, but long-lasting durations are also observed, with overall a broad distribution of contact durations as is typical for data on human interactions[55,61]. The thresholds defined by the tracing policies determine the fraction of these contacts that can be traced by the app. Even slight variations in the tracing policy thresholds may strongly influence the capacity to identify the contacts corresponding to the highest risks of infection, as shown in Fig. 4 by comparing the RSSI and contact duration distributions with the infectiousness curves.

In line with many privacy-preserving contact tracing apps, we additionally assume that each individual device stores the anonymous IDs received from other devices only for a limited time, such that every device does not keep track of all its past contacts but only those of the last $n$ days. This is already implemented in apps used by most countries, applying the privacy-preserving DCT model[28]. We assume $n = 7$ days, and we show in the Supplementary Information (Supplementary Note 3.1) alternative results for shorter and longer tracing memories.

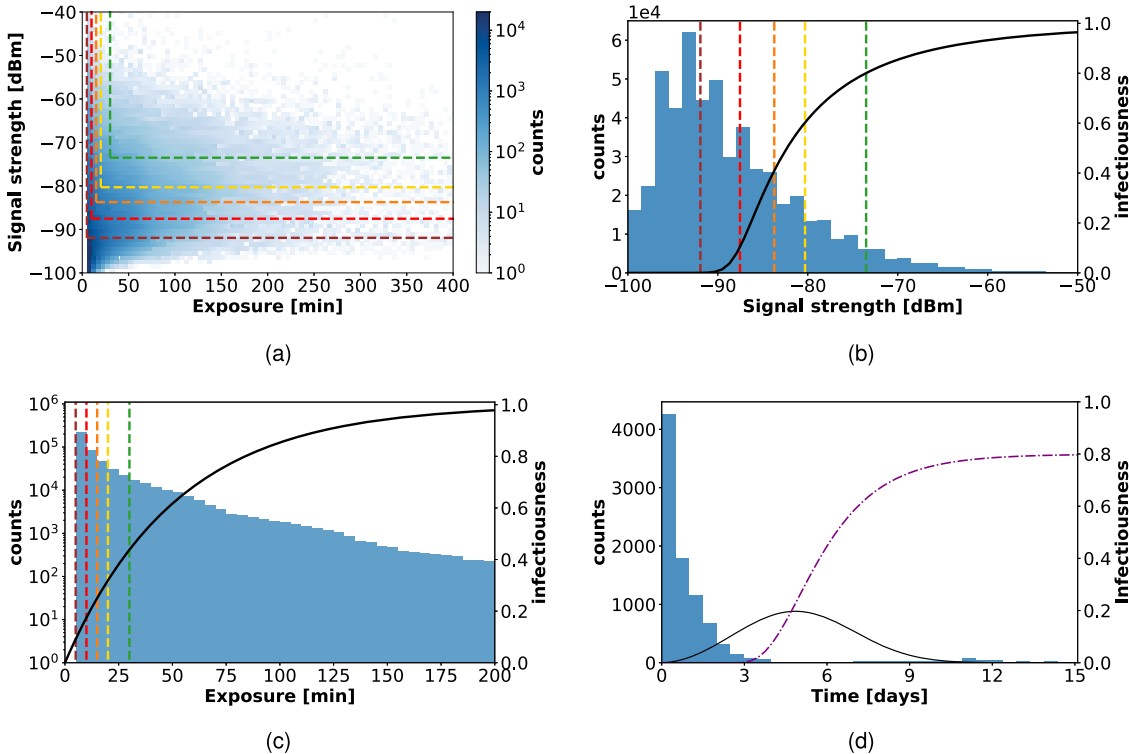

**Fig. 4 Contacts in CNS data set: signal strength, exposure, and inter-contact time.** (**a**): A scatterplot of signal strength vs. duration for all contact events in the CNS data set, displaying the thresholds defining the various policies ($T_p$ for signal strength and $T_d$ for the duration): the contacts identified as "at risk" are those situated above and to the right of the dashed colored lines. (**b**) and (**c**) separately depict the distributions of signal strength and duration, together with the infectiousness functions $\omega_{dist}$ and $\omega_{exposure}$, respectively (black curves), see Supplementary Note 1.2 for their analytical form. (**d**): The distribution of time elapsed between the infection of an individual and their successive contacts, obtained with $\varepsilon_I = 0.8$ and for Policy 5 in the CNS data set. The black curve shows the normalized infectiousness $\omega(\tau)$ as a function of time, and the purple dashed line is the cumulative probability $s(\tau)$ to identify an infected individual.

**Digital tracing enables containment for moderate reproductive numbers**. In this section, we show the results provided by the combination of numerical simulations on empirical data and the theoretical model. The five policies described in Fig. 3 are tested in different scenarios corresponding to different levels of app adoption and different values of $R_0$. Only individuals adopting the app participate to contact tracing; the remaining individuals are outside the reach of the tracing and quarantining policies, but they are still isolated whenever detected because of symptomatic or through random testing. We consider as possible levels of app adoption: 20, 40, 60%. These levels constitute realistic cases, as the fraction of the population that owns a smartphone rarely reaches larger levels (64% for instance for the French population[40,62]), and a certain level of non-compliance should be also considered (from the point of view of the app, non-compliance or non-adoption can be considered as equivalent). As of mid-October 2020, for example, adopters represent 24% of the population in Germany[63,64], 32% in the U.K.[65], and 20% in Italy[66,67].

In addition, each policy is tested with the isolation efficiency values $\varepsilon_I = 0.2$, 0.5, 0.8, 1, which encode isolation capacities ranging from rather poor to perfect isolation of any symptomatic or tested positive person.

The results are shown in Fig. 5. We observe that if $R_0 = 2$, practically none of the policies is able to stop the spreading, even with high app adoption. However, this pessimistic scenario changes under the hypothesis of $R_0 = 1.5$ (second line of panels in Fig. 5), where a larger portion of the phase space implies that the spread can be controlled. An app adoption above 40% is then sufficient to obtain good results: all policies manage to contain the spread for $\varepsilon_I = 0.8$ (except Policy 1 for 40% adoption), and all of them for $\varepsilon_I = 1$.

The situation is even better with $R_0 = 1.2$, as all policies are effective as soon as the isolation efficacy is at least 0.5, even in the case of an app adoption of only 20% (bottom left panel in Fig. 5).

We notice that the tracing efficiency $\varepsilon_T$ varies considerably with different levels of app adoption, but does practically not depend on $R_0$. Indeed, $\varepsilon_T$ only accounts for the fraction of secondary infections that are correctly traced, independently on the spread of the virus and the amount of infected individuals in the population.

The different scenarios explored above draw a framework where $R_0$ is limited by implementing several primary containment measures. DCT is added on top of them and its effect is observed as a component of a broader general effort. While in the absence of DCT a value of $R_0$ larger than one may rapidly lead to a new exponential outbreak and thus to renewed (possibly local) lockdown measures, we have shown here the possible improvement that can be obtained thanks to the deployment of a contact tracing app. The results however highlight that DCT should be accompanied by additional measures and by a sufficient app adoption in order to be effective.

**Any effective containment comes at a cost**. Behind the scenes of the results of the previous section, there is a complex dynamic deserving further investigation. Contact tracing produces in some cases the desirable effect of containing the spread, but side effects emerge as well. Indeed, some of the "at risk" contacts do not actually correspond to a contagion event, while contacts classified as not risky might, as discussed above. It is thus important to quantify the ability of each policy to discriminate between

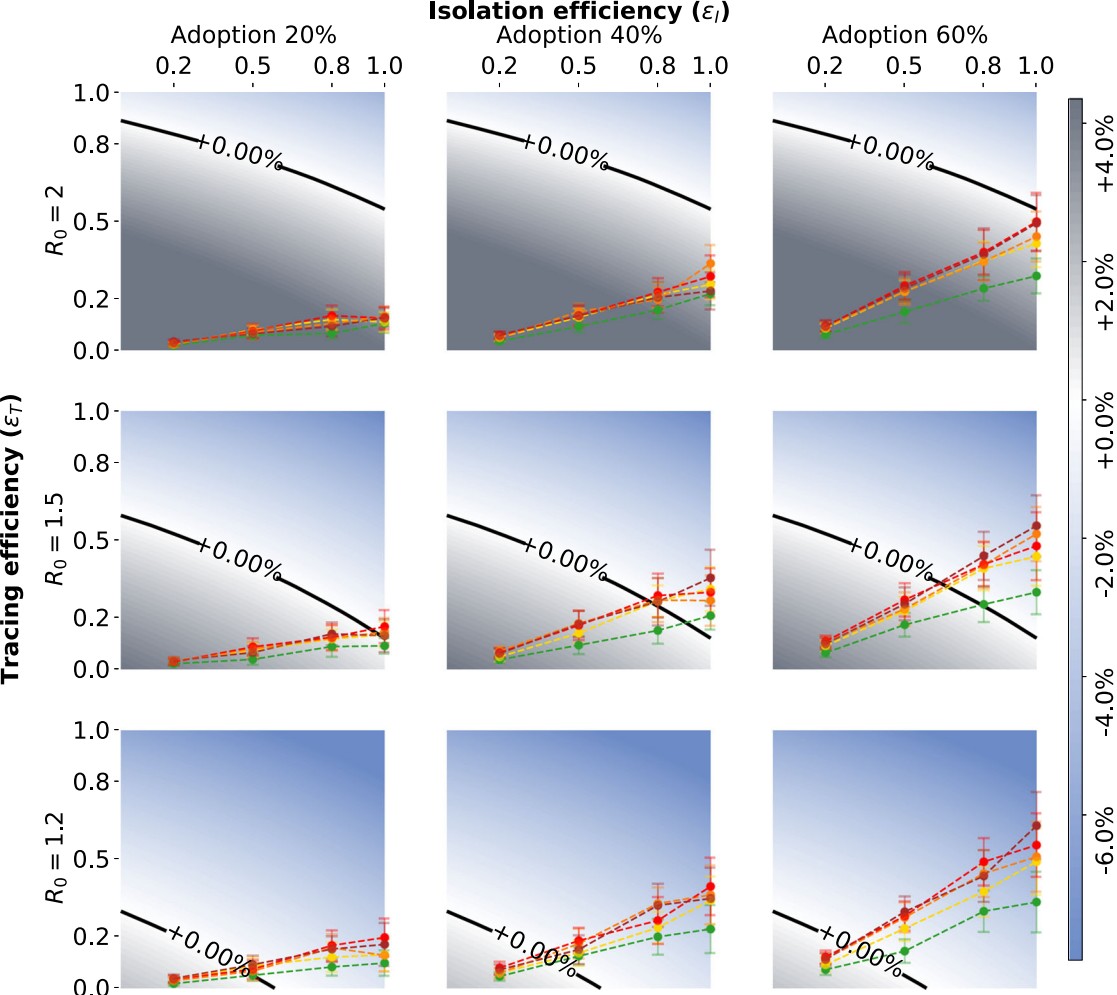

**Fig. 5 Tracing policy efficiency.** Growth or decrease rate of the number of newly infected individuals assuming that symptomatic individuals can be isolated and that an additional 50% of asymptomatics can be identified via randomized testing. The points correspond to the parameter pairs such that the isolation efficiency $\varepsilon_I$ is an input and the tracing efficiency $\varepsilon_T$ an output of the simulations on CNS contact data, for the five policies. The different scenarios are defined by an app adoption level of 20, 40, or 60% (from left to right), and by a value of the reproductive number $R_0$ equal to 2, 1.5, or 1.2 (from top to bottom). All the points have been obtained as mean values over $n = 200$ simulations and the error bars represent the standard error.

contacts on which the disease actually propagated and the others, in terms of false positives (quarantined individuals who were not infected) and false negatives (non-quarantined infected individuals). To visualize this behavior, we focus on the setting with $R_0 = 1.5$ and $\varepsilon_I = 0.8$, with an app adoption of 40%, since it is representative of a situation in which some policies are effective in containing the spread and others are not (see Fig. 5, center). The corresponding time evolution of the average percentages of false negatives and of false positives over the population for each policy are shown in Fig. 6.

In terms of epidemic containment, the best policies are those that can rapidly reduce the number of active infected, i.e., of false negatives. In the case of Policy 1, this number remains quite high for the entire simulation time, whereas for all other policies the number of false negatives remains lower. These policies lead overall to a larger value of the tracing effectiveness $\varepsilon_T$ (see Section "Methods"), thus leading to a better epidemic containment.

The smaller number of false negatives for the effective policies comes however at the cost of an increased number of false positives, as shown in Fig. 6b. In other words, as a policy becomes more effective in tracing actually infected individuals, it also leads to the quarantine of individuals that have not been infected but that had a contact classified as risky by the tracing policy. The

maximal number of false positives is very sensitive to the specific policy, contrarily to the number of false negatives. In particular, it appears from the analysis of the previous Section, "Digital tracing enables containment for moderate reproductive numbers", that Policies 2, 3, 4, and 5 have a similar effectiveness to contain the epidemic and Fig. 6a shows that they yield similar numbers of false negatives, but their undesired side costs are different, as the broader definition of risky contacts produces a larger number of false positives. This highlights once more the importance of the fine-tuning of the chosen policy. Since balancing between these two effects may be non trivial, we plot in Fig. 6c the effectiveness vs. cost for each policy, showing that Policy 2 is favorable in that it achieves an almost maximal effectivity (small number of false negatives) at a very low cost (small number of quarantines). Figure 6d reports the average percentage of the population that had to quarantine in the simulations (increasing from policy 1 to 5) and the percentage of those were actually infected (decreasing from policy 1 to 5).

To further facilitate the challenge of choosing the right policy, in Supplementary Note 3 we test the behavior of the model under extended scenarios to precisely quantify the sensitivity of the outcomes with respect to changes in our fundamental assumptions. The model robustness is assessed by changing the tracing

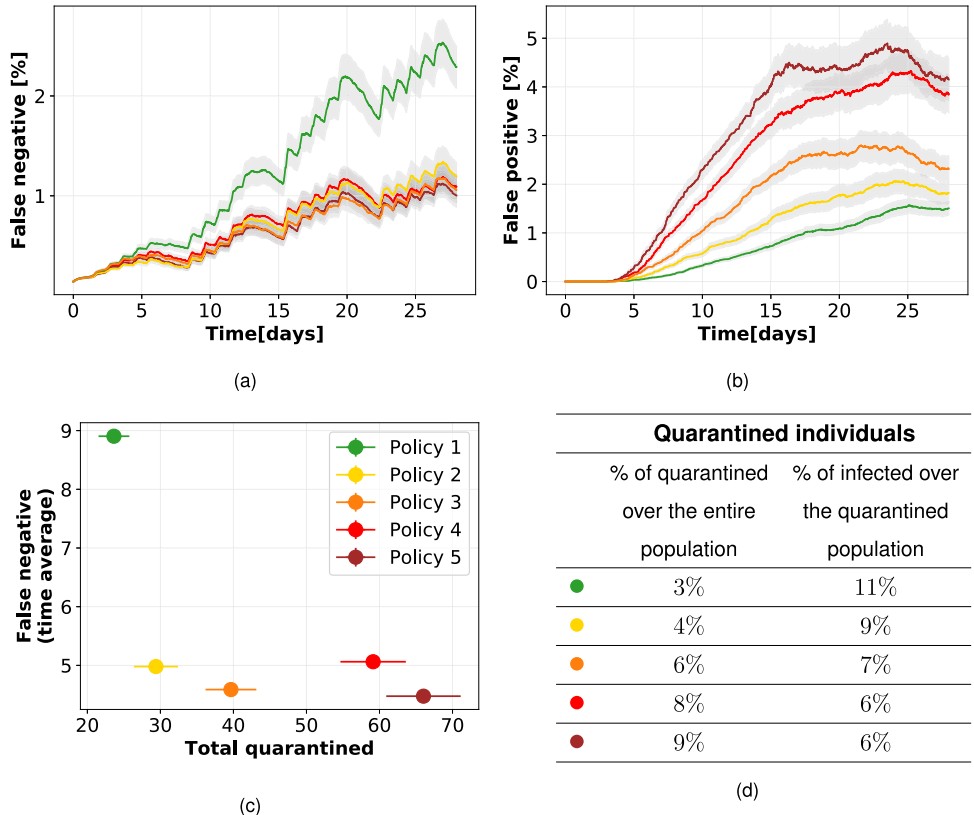

**Fig. 6 Quarantines, false positives, and negatives, with 40% app adoption and $R_0 = 1.5$.** Temporal evolution of percentages of false negatives (**a**), i.e., infected individuals not quarantined, and false positives (**b**), i.e., not infected individuals quarantined, over the population for the five different policies, assuming an isolation efficiency of $\varepsilon_I = 0.8$. The graphs depict the mean and standard error over 200 independent runs. (**c**): Effectiveness (low number of false negatives) vs. cost (total quarantines) of the policies. (**d**): The table reports the percentage of distinct individuals who have been quarantined over the entire population and the percentage of them who were actually infected (true positive).

memory (longer and shorter) in Supplementary Note 3.1, the reporting delay in Supplementary Note 3.2, the ability to trace second-order contacts in Supplementary Note 3.3, the fraction of asymptomatic infected in Supplementary Note 3.4, the adoption of modified policy thresholds in Supplementary Note 3.5, and a different response of the population to the request of multiple quarantines in Supplementary Note 3.6.

## Discussion

**Policies for DCT: implications and constraints**. In the modeling of contact tracing, considering several scenarios of isolation efficiency, app adoption, and $R_0$ values is of foremost importance in order to account for the complex and heterogeneous issues connected with concrete policy implementations.

These issues should be clear to any policy maker having to decide on containment measures, in order to understand that contact tracing is a viable containment strategy for COVID-19 only in conjunction with complementary policies, as the results of the previous sections show.

These considerations enter our modeling approach in several ways. On the one hand, some parameters are related to the healthcare system capacity and to the socioeconomic condition of the population. These include the isolation efficiency $\varepsilon_I$ and the delay in the case reporting, which should account for potential heterogeneities in the access to tests and in the possibility of a person to isolate. This last involves in particular both the access to appropriate spaces and the economic feasibility of a temporary cessation of the working activity. Since each country has a different level of capacity to isolate individuals we considered several levels of $\varepsilon_I$ instead of prescribing a fixed setting. The delays

in turn depend on factors of different nature such as the delay in reporting, the availability and response of the call centers and of the health authorities, the app- and app-backend- related delays, etc. The analyses reported here take into account a delay of 2 days in isolating infected cases (thus in tracing and quarantining their contacts). This realistic delay does not prevent the proposed policies from keeping the epidemic under control, which is possible under some conditions. However, we observe that a larger delay, even if only one additional day, leads to a completely different scenario (reported in Supplementary Note 3.2) where assuming $R_0 = 1.5$ and 40% app adoption, none of the proposed policies proves able to contain the epidemic, even for maximal isolation efficiency, and despite the higher numbers of quarantines, false positives and false negatives.

Moreover, we have analyzed the effect of the app within epidemic scenarios of limited reproductive numbers ($R_0 = 1.2, 1.5, 2.0$), which are the result of the implementation of complementary policies in addition to DCT. Such measures include traditional manual tracing, mask use, and physical distancing.

Our model also includes the level of app adoption as an explicit parameter and we consider 20, 40, and 60%. It should be taken into account that factors like the limited access to supported smartphones for different age and income brackets, but also the willingness to adopt the app (strongly dependent on people's trust in DCT and health system), are crucial elements that contribute to these values.

All these parameters should be set with some care. The design of our model allows us to treat them as tunable inputs and in particular, no unrealistic or idealized assumption on these parameters needs to be made.

Privacy issues raised by digital tracing are also of great importance, and they have been extensively discussed[35,68–70]. For these matters we refer to the decentralized models that have been developed such as the Decentralized Privacy-Preserving Proximity Tracing (DP-3T)[28], and to the discussion therein. In particular, we adopt a tracing scheme that does not need to access the complete network of contacts at any time but is based only on decentralized exchange of anonymized keys.

**DCT: insights and limitations**. The general model that we developed for studying the effect of isolation and contact tracing on controlling the COVID-19 epidemic is inspired by the work of Fraser et al.[42]. The main distinctive characteristics that we have introduced are the following: (i) a general mathematical model that allows to evaluate the evolution of an epidemic in the presence of isolation and DCT at finite time; (ii) the evaluation of tracing efficiency by means of a numerical simulation on real contact data, and no more as an arbitrary parameter of the model; (iii) the dependence of infectiousness on the actual duration and physical proximity of contacts; and (iv) consequently, the design of appropriate policies.

The functional shape of the infectiousness that we devised is composed by three dependencies: the time since primary infection $\omega(\tau)$, the duration of a contact $\omega_{\text{exposure}}(e)$, and its proximity $\omega_{\text{dist}}(s_s)$. The first is originally suggested by Ferretti et al.[17], while the other two were introduced in this work. We have shown that the implemented model is robust to changes of all three contributions, see Supplementary Notes 1.2 and 1.3.

Our results suggest that an insufficient app adoption may render any digital tracing effort helpless on its own, if the reproductive number is too high. In view of these results, bridging the gap between a realistic app adoption and the larger tracing capability required to contain the disease appears crucial. This goal can only be reached with a joint effort of policy makers and health authorities in organizing an effective manual tracing, and of individual citizens in adopting the app. We therefore tested different levels of app adoption and a range of possible values of $R_0$, reduced from its original value by other restrictive measures, like masks wearing and physical distancing.

Moreover, we found that the set of parameters that allow containment of the spread is strongly influenced by the fraction of asymptomatic cases. By first assuming an ideal setting where any pair of parameters $\varepsilon_I$, $\varepsilon_T$ is possible, we showed (Fig. 1) that the area of the phase space representing the setting where it is possible to control the epidemic is reduced when considering 20% or, worst-case scenario, 40% of asymptomatic individuals in the population.

We tested five policies to define risky contacts that should be traced (Fig. 3), with different restriction levels. Our results highlight how isolation and tracing come at a price, and allow us to quantify this cost using real data: the policies that are able to contain the pandemic have the drawback that healthy persons are unnecessarily quarantined. In other words, achieving a rapid containment and a low number of false negatives requires accepting a high number of false positives. This stresses the importance of a fine tuning of the tracing and isolation policies, in terms of the definition of what represents a risky contact, to contain the social cost of quarantines. Let us observe that this last could be mitigated by testing the quarantined population and revealing the false negatives, thus translating the social cost in an economical burden due to swabs. Among the tested policies, those that appear to provide the best balance between effectiveness and cost are Policies 2 and 3, corresponding to considering as risky a contact longer than, respectively, 20 and 15 min, with distance shorter than, respectively, around 2 and around 3 m.

This is in agreement with the European guidelines for high-risk contacts[71].

We modeled the tracing procedure assuming that contacts are stored in each user's app for 7 days. Such tracing memory seems a good balance between the too short 2 days, which fails in containing the epidemic, and the too long 15 days, expensive in terms of quarantines and not leading to strong improvements in the spread containment (Supplementary Note 3.1).

We also included in our model a delay of 2 days in isolating the infected individuals. This delay might however increase when the number of infected cases grows. For this reason, we tested a delay of 3 days too, revealing a much worse scenario (Supplementary Note 3.2). This highlights the importance of readiness in implementing the testing and isolation procedure, as increased delays might neutralize the beneficial effects of the app.

Another important result concerns the issue of privacy: we numerically tested a second-order tracing, where also contacts of an infected individual are quarantined. Such procedure leads to a strongly enhanced risk in terms of privacy, but we found that it determines a useless massive quarantine while failing to bring any clear beneficial effect on controlling the epidemic (Supplementary Note 3.3).

Finally, we tested the possibility that people reduce their compliance if they are notified multiple times and asked to quarantine despite not being infected. This might indeed lead to some mistrust in the DCT procedure and in the healthcare and government institutions. The results that we obtain are very similar to those found with the standard procedure, where the level of compliance is set at the beginning and does not depend on the multiplicity of quarantines. This further confirms the robustness of our general model and of our results (Supplementary Note 3.6).

Our study comes with a number of limitations. First, we have considered data corresponding to a few limited social environments (a university campus, a high school, and a workplace) and we cannot provide an overall general study that includes multiple and differentiated contexts and their mutual interplay. Moreover in each data set, only people involved in the experiment have been tracked, neglecting other contacts occurring outside their school, university campus or workplace. Hence, the complete data sets only provide access to part of the interactions of the involved individuals, which is useful to analyze contact tracing in specific environments but does not provide a full picture of a society, e.g., an entire city. This limitation is due to the current lack of larger data sets involving people belonging to different environments, which would represent the general interactions within the population of a city or a larger geographical area. In addition, the implemented policies have been necessarily tailored to the specific CNS data set, depending on the available values of RSSI supported by the used smartphones. Those might differ in actual implementations of DCT apps currently in use in different countries, probably relying on a more advanced technology. Nevertheless, we emphasize that even if we used the simulations performed on these data sets to obtain a realistic quantification of the tracing ability, the controllability of the disease is itself assessed by the general mathematical framework. The results that we present are hence general, not bounded by specific data sets, but only numerically supported by real data to have a realistic implementation of tracing.

Moreover, our study is limited by the current knowledge of the contagion modalities of the SARS-CoV-2 virus, in particular concerning its dependence on the physical distance among people and the duration of their contacts. The curve of infectiousness has been designed based on previous contagion studies and on reasonable assumptions (also considering reduced transmissibility of asymptomatic people). Additional refinements of the transmission dynamic could be obtained by accounting for aerosol

transmission, adding a dependence from the environment characteristics, such as being indoors or outdoors, and the presence or not of ventilation. This factor could in principle be modeled by considering information on the (co-)location of the individuals, which is available for some SocioPatterns data sets[47]. Should new insights emerge in the way the virus spreads, these could be easily incorporated into our model.

Finally, we model delays in the case reporting and thus in the isolation process, but assume that the quarantine notification of the traced contacts is instantaneous. This is reasonable and it is one of the advantages of relying on DCT, but two factors may introduce a delay: the app may check for at-risk exposures only 3–4 times a day, and the backend servers that distribute "infected" keys to the app often batches them before notification. The combination of these factors introduces an average delay of several hours (4–5 h) and a worst-case delay of half a day.

Despite these limitations, the presented model represents an important contribution to the discussion about DCT, proposing a refined approach that allows to investigate a number of features that are unattainable with other recent models (see Supplementary Note 6 for a discussion of the state of the art models of DCT).

In conclusion, this combination of a well-established epidemic model with state-of-the-art, empirical interaction data collected via radio-based proximity-sensing methods, allows us to understand the role played by intrinsic limitations of digital tracing efforts, affording a viewpoint on the ambition of achieving containment with digital interventions. Namely, we are able to test and quantify the role that a real contact network plays both for the infectiousness of contact and for the ability of a policy to detect it and to respond optimally.

## Methods

The algorithm modeling the spreading and containment of the virus is implemented on the real contact network and coupled with the mathematical model.

This simulation is used in two ways. First, it produces results that are averaged over the network and then aggregated into a quantity, $\varepsilon_T$, that can be plugged into the mathematical model. In this step, the network simulation is used as an estimator of a real-world parameter value. We remark in particular that the prediction of the outcome of the policies (epidemic containment or exponential contagion) is obtained solely from the mathematical model, informed with these real-world parameters.

On the other hand, the simulation on the network goes beyond the mathematical model in that it captures complex and non-uniform events and the heterogeneity of individual behaviors. The simulations thus give also access to several fine-grained quantities of interest that provide a complementary view on the epidemic. In particular, we can measure the number and time evolution of false and true positives, offering a quantification of the cost of the quarantine measures.

In the following, we detail the implementation of the numerical simulations (Section "Spreading and tracing on the real network") and the methods used to extract the aggregated parameters (Section "Aggregation and parameter estimation").

**Spreading and tracing on the real network**. The contact data set is represented as a temporal sequence of undirected and weighted graphs. The nodes of the graphs are the individuals stored via their unique identifiers, and an edge connects two of them if their respective Bluetooth devices have recorded each other. The weight of each edge is the pair of the signal strength and the duration of this contact. These two values are obtained by aggregating the continuous measures of the data set on successive time windows of duration 300 s.

The simulation keeps track of the status of each node, which is updated depending on the spread of the infection (which is a stochastic phenomenon regulated by the infection probability $\omega_{data}$) and on the enforcement of the tracing and isolation policy (which is again stochastic, and dependent on the definition of the policy's thresholds).

The simulation is parametrized by two types of inputs: disease-dependent parameters, which are discussed in Section "A modeling framework for DCT on empirical contact networks" and Supplementary Table 2, and tracing-dependent parameters, which are the isolation efficiency $\varepsilon_I \in [0, 1]$, the memory length of the contact tracing, the duration of the quarantine, and the fraction of app adopters in the population.

Once these parameters are set the algorithm works as follows:

- *Setup*: A fraction of the nodes, extracted uniformly at random, is set to non-adopters, i.e., not using the app. They will contribute to the spread of the virus and they can be isolated, but their contacts cannot be traced and they cannot be quarantined. Observe that we make the simplifying assumption that the app influences only the quarantining of individuals, but not the isolation policy. Namely, we assume to be able to detect and thus isolate an infected individual independently of the app, while we are able to trace the contacts only between pairs of app adopters.
- *Initialization*: A randomly extracted node from the first graph of the sequence is set to infected. It is assigned a time since infection chosen uniformly at random in [0, 10] days.
- *Time evolution*: For each temporal step the following steps are repeated:
- *Update contacts*: The list of contacts of each app adopter node is updated by adding the contacts of other app adopters at the current time, if they fall within the policy's thresholds. Each list stores the contacts for a fixed maximum number of days (which is set to 7 days in the main simulations).
- *Update quarantined*: The list of quarantined nodes is scanned. Nodes who completed the quarantine time (10 days in the main simulations) are just removed from the list if healthy, or removed and added to the list of isolated if they developed symptoms.
- *Update infected*: The list of infected nodes is scanned. Those who became symptomatic or are tested positive, depending on the probability onset_time (·) (see Section "A modeling framework for DCT on empirical contact networks" and Supplementary Table 2) are added to the list of infected identified by the health authority. Then, the list of identified infected is scanned, and each of its nodes is isolated with a probability $\varepsilon_I$. For each successfully isolated node that is an app adopter, the tracing policy is enforced on its contacts, i.e., all the nodes registered as contacts are quarantined. All the other infected nodes instead can spread the infection: each of their neighbors is infected independently with a probability modeled by $\omega_{data}$ (Supplementary Note 1.2).
- *Check quarantined*: The list of quarantined is scanned again to find symptomatic nodes. If a symptomatic node is found, it is isolated and the tracing policy is enforced on its contacts who are app adopters.

Observe that the contacts taken into account for the contact tracing are defined according to a given policy's thresholds (distance and duration), i.e., only those interactions with sufficient duration and small enough distance are stored in the contact lists. However, the spreading process can a priori occur between an infected node and any of its neighbors, the probability of a contagion event being given by $\omega_{data}$.

Moreover, the simulation assumes that each individual that is required to quarantine is willing to do so. We consider in Supplementary Note 3.6 the situation where individuals have a decreasing acceptance to comply, based on the number of times that they are asked to quarantine. On the other hand, the compliance to isolation is already modeled by the user-defined parameter $\varepsilon_I$, which represents the effective fraction of identified infected who successfully isolate, where the value of this fraction may depend on the health system capacity, but also on the nodes' compliance and the possibility to isolate.

**Aggregation and parameter estimation**. During the simulation, whenever the tracing and quarantine policy is enforced a quarantine error $e_T$ is computed to score its success. This value is defined for each isolated node as the ratio between the number of its secondary infections (i.e., the nodes that it infected) that did not quarantine, and the total number of its secondary infections.

The list of values $e_T$ (one for each isolated individual) is collected and averaged over the entire simulation to obtain a mean score $\langle e_T \rangle$. This value encodes the contributions of the chosen policy, of the adoption rate, of the duration of the memory of contacts, and in general of the heterogeneity of the network dynamics.

This allows to assign to each policy a tracing efficiency $\varepsilon_T$ observed over the simulation as a function of its inputs and of the network dynamics. We define it as the product of two independent factors modeling the efficiency of the isolation (individuals who are not isolated are automatically excluded from the contact tracing, so their contacts do not quarantine) and the effect of the quarantine error, as:

$$\varepsilon_T = \varepsilon_I (1 - \langle e_T \rangle). \tag{2}$$

A perfect efficiency of the tracing policy ($\varepsilon_T = 1$) is possible only under perfect isolation ($\varepsilon_I = 1$) and zero quarantine error ($\langle e_T \rangle = 0$).

Considering $\varepsilon_I$ as a free parameter allows us to explore different scenarios, thus providing a full range of predictions. This choice accounts for the fact that in a realistic scenario the ability to identify and consequently isolate an infected individual is set by the number of tests that are implemented and by their accuracy, features whose identification is out of the scope of this work. We mention that the adoption of an app might have a positive effect on this quantity if the possibility of self-reporting when symptoms appear is implemented in the device.

**Reporting summary**. Further information on research design is available in the Nature Research Reporting Summary linked to this article.

## Data availability

The data that support the findings of this study are publicly available. The CNS data can be found at https://doi.org/10.6084/m9.figshare.7267433and the SocioPatterns data at http://www.sociopatterns.org

## Code availability

We are pleased to make available the source code accompanying this research[72]. The code uses Python (version 3.8.3), Numpy (version 1.18.5), Scipy (version 1.2.0), Networkx (version 2.5), Matplotlib (version 3.0.2).

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

## Acknowledgements

The authors would like to thank Esteban Moro, Alex Sandy Pentland, and Fabio Pianesi for early discussions and useful comments, Stefano Merler for the feedback on the design of the infectiousness parameters for COVID-19, and Valentina Marziano, Lorenzo Lucchini, and Luisa Andreis for the discussion and general support. This study was partially supported by the ANR project DATAREDUX (ANR-19-CE46-0008-01) to A.B. C.C. acknowledges partial support from the Lagrange Project of ISI Foundation funded by CRT Foundation, and from the EU Horizon 2020 grants EPIPOSE (SC1-PHE-CORONAVIRUS-2020) and PERISCOPE (SC1-PHE-CORONAVIRUS-2020-2C).

## Author contributions

G.C, G.S., and B.L. conceived the idea. G.C., G.S., A.L., and E.P performed the analytical calculations and numerical computations. All the authors contributed to research design, analytical development, critical revisions, and wrote the paper.

## Competing interests

The authors declare no competing interests.
