## [Peer Review File · Nature Communications]

Reviewers' Comments:

Reviewer #1:

Remarks to the Author:

This is an interesting and original manuscript which addresses important issues related to the current COVID-19 pandemic.

And I wish to thank the authors for writing a so easily readable and understandable manuscript.

Their contributions are 3 fold. 1) They extend an existing modeling framework informed by high-resolution contact data. 2) Based on this framework they develop different tracing strategies, and 3) they dig into what costs, in terms of false positives and negatives, each strategy has.

Their work provides considerable improvement of the studies done by Fraser et al. and Ferretti et al. with the authors developing a model that requires no assumption of the functional form of the contagion (growing or decreasing).

A further strength of their approach is that parameters ϵ_I and ϵ_T no longer need to be assumed as independent.

I think this manuscript would be a good contribution to Nature Communication, however, I only recommend it to be accepted after the below shortcomings are addressed.

The manuscript is very technology centric. The authors present digital contact tracing as a techno-utopia but leave out broader societal discussions of inequality, discrimination, and power. Furthermore, they neglect to discuss the many privacy and ethical issues which have been raised by multiple researchers around contact tracing apps. For examples see below:

- <https://www.nature.com/articles/d41586-020-01578-0>
- https://www.scss.tcd.ie/Doug.Leith/pubs/opentrace_privacy.pdf
- https://www.scss.tcd.ie/Doug.Leith/pubs/contact_tracing_app_traffic.pdf
- <https://www.amnesty.org/en/latest/news/2020/06/bahrain-kuwait-norway-contact-tracing-apps-danger-for-privacy/>

I believe the authors have an obligation to mention these issues, because governments and public health agencies might read this manuscript as a validation that contact tracing is effective, when in fact, in most cases it is not a viable solution to the pandemic - as some of the results from this manuscript also show (although this could be discussed more critically).

Below, I have split my comments up into two sections, major and minor issues.

Additionally it might be beneficial for the public discussion around digital contact tracing if the authors, based on their results, build a decision matrix which can be used by government to decide if/when to roll out contact tracing apps.

I.e. depending on factors such as adoption rate, R_0 , testing rates, etc.

I will let it up to the authors to decide if this is something they want to pursue.

Major issues:

-- 1) If we start with the Manuscript title, it is very broad: "Digital Proximity Tracing in the COVID-19 Pandemic on Empirical Contact Networks". However, the authors only focus on a very specific part of the current COVID-19 pandemic - namely the efficacy of digital contact tracing in the case of re-opening of societies, or places with relatively low infection rates (low R_0).

The title should reflect these limitations, for instance the title could potentially be: "Digital Proximity Tracing in the COVID-19 Pandemic on Empirical Contact Networks: Controlling re-emerging outbreaks"

-- 2) The authors use very optimistic estimates of adoption rates and smartphone ownership. For digital contact tracing to work, people need access to smartphones and Internet connectivity. However, access to technology is not uniformly distributed across different layers of society. In fact, studies have shown there are wealth-, age-, gender-, ability-, and education-gaps in smartphone ownership.

If we look at country specific statistics, in some countries smartphone ownership is below 25%, meaning fewer than 1 in 4 people own a smartphone (see more here 

<https://www.pewresearch.org/global/2019/02/05/smartphone-ownership-is-growing-rapidly-around-the-world-but-not-always-equally/>).

Further, most of the decentralized contact tracing app-frameworks, which the authors describe, rely on BLE technology, which is not available in approx. 2 billion of existing smartphones - predominantly because they are of an older make (read more here 

<https://arstechnica.com/tech-policy/2020/04/2-billion-phones-cannot-use-google-and-apple-contact-tracing-tech/>).

All these issues compound and globally bring down the percentage of individuals who own a smartphone suitable for contact tracing to below 50% of the world population

(https://www.gsma.com/mobileeconomy/wp-content/uploads/2020/03/GSMA_MobileEconomy2020_Global.pdf).

If we look past smartphone adoption rates and instead focus on whether people are willing to install and use the apps things look even more dire.

In most countries only a small fraction of the populations have installed and use national tracing apps (The most up-to-date estimates of adoption rates, that I'm aware of, can be found here 

https://docs.google.com/spreadsheets/d/1_BCKIMuniEhzvpQ-ha0jhdksvqdINUUAUHA8J9LSr_Dc/edit#gid=0).

Further, a recent survey for the US shows that 7 out of 10 individuals say they do not even want to install tracing apps (<https://arstechnica.com/science/2020/06/more-than-7-in-10-americans-dont-want-contact-tracing-data-shows/>).

Even in a country like Germany, with relatively high trust in the government, the national tracing app has only been installed 13 million times (not to be confused with it being installed by 13 million people), compared to the total population of 83 million, this amounts to a very low ~16% adoption rate.

All these factors compound to lower the adoption rate of contact tracing apps.

While the authors consider different adoption rates and acknowledge that "high level of app adoption is crucial to make digital contact tracing an effective measure", their most conservative estimate of 60% is very far from reality.

To have an open and unbiased discussion around the usefulness of contact tracing the authors need to include adoption rates of 20% and 40% in their study. Fig 4 would be a good place to depict the effect of low-adoption rates. For instance, by adding two additional columns to the figure.

-- 3) For contact tracing apps to work there needs to be access to equitable testing. In the manuscript the authors indirectly assume this, but never explicitly mention that without widespread and equitable testing contact tracing frameworks wont work.

This ties in with one of the key parameters of the model, $s(\tau)$ the probability for an infected individual to be recognized as infected within a period of time τ , via some form of testing regiment. As a value of τ the authors use 2 days, which in a "normal" setting seems like a reasonable choice.

However, experiences from different countries show that this is not the case. In the US, for instance recently released statistics show that the average turnaround time for a test is 7 days (<https://www.cnn.com/2020/07/13/us-coronavirus-surge-leads-to-testing-delays-across-the->

nation-quest-diagnostics-says.html).

How robust are the results to settings in which cases are reported with a delay of more than 2 days?

-- 4) The work builds on the assumption that individuals:

- 1) are rational agents who will optimize common good through self-quarantine once notified of exposure
- 2) have access to a safety net in order to properly quarantine

While I was glad to see that the authors try to model how many people can self-isolate through the parameter ϵ_I . This parameter will change (and probably decay over time) if individuals are notified multiple times. Basically people will reach a level of fatigue and stop following the notifications from the app, once they have been notified for the n 'th time.

Only the most well-off individuals and people living in societies with social and labour safety nets will be able to properly follow the recommendations from the apps. For instance, we know that people working in the most vulnerable jobs (cashiers, bus-drivers, teachers) will be notified more frequently than the average person, but they will not necessarily have the means to self-quarantine.

While a comprehensive study of this is out of the scope for this manuscript, I would like to see statistics for how many times individuals are asked to self-isolate.

E.g. run the simulations over 50 days, with a self-quarantine period of 14 days, what will be the frequency distribution of exposures?

Put differently, out of the 706 participants from the Copenhagen Network Study how many of the participants will receive the notification once, twice, thrice, etc.?

-- 5) Comparing tracing strategies to full lockdowns.

In Figure 3 the authors show, that depending on the tracing strategy, thousands to tens of thousands of contacts could be flagged as contagion events. Since the study population is relatively small (706 individuals for the Copenhagen Network Study, and different for the other datasets), I would like to see how large a fraction of the full population each strategy would require to be in quarantine over time. This will enable the reader to compare digital tracing to a full lockdown.

To a certain degree the authors try to show this in Fig 5, however, they never show how large a fraction of the body of participants that would mean.

Judging from the table in Fig 5 the results seem to suggest that policies 4 and 5 require around 40-60% of the study participants (population) to be in quarantine - which is not very different from a full lockdown. The only difference is that this "hybrid lockdown" with people going in and out of quarantine would be a logistical nightmare.

Can you please do a plot, comparing the 5 strategies, with time on the x-axis (0-50 days), and the cumulative proportion of population that has been in lockdown on the y-axis?

-- 6) Repeatability and context of contacts. Students in the Copenhagen Network Study will most likely interact with the same individuals across time, due to heterogeneities created by class structures, studylines, seniority of students, etc.

As such individuals will most likely be exposed to contagion events within these structures.

How effective are the five strategies at identifying contagion events across these heterogeneities?

I.e. how large a fraction of the events depicted in Fig 3 can be attributed to within-group vs between-group?

-- 7) There is no discussion around masks, which I find very disturbing.

In their results the authors show that digital contact tracing works when, and only when, the following conditions are met:

- 1) high smartphone ownership and app adoption, preferably higher than 60% of population - which no country has achieved so far
- 2) randomized mass testing schemes, to enable that many asymptomatic carriers are discovered and quarantined - very few places have the resources to do this
- 3) a social safety net where people have the resources to self-isolate when notified of exposure - very rare outside of Europe (even in Europe, particularly for the most vulnerable populations)
- 4) high tracing and isolation efficiencies (ϵ_T and ϵ_I)
- 5) Limited spread of COVID, i.e. Low reproduction numbers (low R_0 values)

Based on these assumptions the authors conclude that digital contact tracing can be used as another non-pharmaceutical intervention to reduce the spread of COVID.

I think this is a fallacious conclusion, because advocating for deploying promising but untested technologies without mentioning masks is deceiving.

Masks work to stop the spread of COVID no matter whether people have smartphones or not, or whether R_0 is small enough, or whether there is enough mass testing.

The authors need to put the "effectiveness" of digital contact tracing into perspective to the efficacy of low-tech solutions such as masks.

And they also need to be more critical in discussing their results.

For more info about low-tech solutions that authors can read more here 

[https://www.thelancet.com/journals/lancet/article/PIIS0140-6736\(20\)31142-9/fulltext](https://www.thelancet.com/journals/lancet/article/PIIS0140-6736(20)31142-9/fulltext).

-- 8) Technical Issues.

The assumption of the authors policies builds upon a dataset where all users have identical devices (smartphones in the case of the Copenhagen Network Study or RFID tags in the case of the supplementary datasets), but this is not true in real-world settings. How well do the authors think this will work for a setting where individuals have different smartphone brands?

-- 9) The proposed model and corresponding code provides a great testbed to test additional scenarios and contact networks. Will the authors open source the model code?

Minor issues:

1. Using the term "social distancing" sends the wrong message. While "social distancing" was initially widely used the conversation has shifted towards "physical distancing".

This is reflected in the WHO's guidelines (<https://www.who.int/westernpacific/emergencies/covid-19/information/physical-distancing>) which also say that while people need to physically distance they should stay socially connected.

Please update your manuscript to reflect this.

2. You are inconsistent in how you write lockdown and use both "lock-down" and "lockdown" in the text. I suggest you change lock-down(s) -> lockdown(s) on l.17, l.38, l.46, l.48, l.131, l.496, and l.953

3. l.38 prove -> proven to be

4. l.152 covid-19 -> COVID-19 (to be consistent in how you refer to the virus in other places in the MS and also to align with the official name of the disease)
Same on l.170, l.242, l.339
5. l.160 his/her -> their
6. l.161 s/he -> they
7. You never reference Fig 1 in the manuscript
8. l.197 "below the thresholds" -> "below an RSSI threshold"
9. I'm nitpicking here, but it might make sense to switch the ordering of figures 2 and 3, to reflect the order you refer to them in the text.
10. On l.449 you say the the Copenhagen Network Dataset contains contact information for 1 month, however, on l.225, you model the infection up until $T = 50$ days. How?
If this is because you do not use contact networks for the idealized scenario it would be good to indicate this directly in the text.
11. l.255, you are referring to a Table in Figure 4, but Fig 4 has no table.
12. Fig 4, could you please add the respective colorbars for each of the subplots, or just one for all of the plots?
13. Results of simulations in Fig 4. I am assuming you have done multiple simulations (since they tend to be stochastic) and averaged over them, could you please indicate the errorbars for the individual points in the Figure.
14. Fig 5, the false negative and false positive plots, what are the y-axes showing? Is this infection events (edges) or infected individuals (nodes)?
15. L.321-323. The sentence "For all other policies instead the curves of false negatives all reach similar levels" says that policies 2-5 all have similar levels of false negatives. However, the next sentence then says that "The curve drops to zero rapidly however only for the stricter policies, while for Policy 2 it remains at a higher level, ..." meaning that Policy 2 is different from Policies 3-5. These two sentences are confusing and inconsistent, you cannot argue that Policies 2-5 are the same, and then say that Policy 2 is different from Policies 3-5.
16. l.464, the weighted network you are constructing is not necessarily represented by a distance weighted connections. There are lots of issues with interpreting RSSI as physical distance in complex environments (lots of concrete, metal, etc), see more here  <https://arxiv.org/pdf/2006.08543.pdf>. Instead your edge weights stand for "temporal and signal strength-weighted connections".
17. On l.467 you are saying that simulations start from an initial number of infected people, Y_i , however, you never mention how Y_i is chosen/sampled

Reviewer #2:

Remarks to the Author:

In the manuscript titled “Digital Proximity Tracing in the COVID-19 Pandemic on Empirical Contact Networks”, the authors utilize mathematical models to explore the feasibility to achieve SARS-CoV-2 control based on digital contact tracing app, utilizing proximity sensors commonly available in smartphones. The topic of the manuscript is interesting and relevant to the current situations of the SARS-CoV-2 pandemic, as many countries and public health agencies have developed and deployed digital tracing apps to assist epidemiological investigation and control. The key contribution of the study is the utilization of interaction data collected by the “Copenhagen Network Study” prior to the COVID-19 pandemic, which is also collected through smartphones. The dataset provides a very unique perspective on the spatiotemporal granularities of the interaction data that a commonly available smartphone today is capable of capturing and cannot be obtained through traditional epidemiological investigation. The manuscript has the potential to make an important advancement in this particular area of research. However, there are a few major issues that need to be addressed before it can be considered to be published at Nature communication.

First, improvement in writing is recommended for the authors before the manuscript could be considered for publication. It’s understandable that during the pandemic era the authors are challenged to put together a rather complex study given limited time. However, the manuscript at current stage is quite difficult to follow overall. Specifically:

- The “Introduction” section could be shortened (3 pages currently) with a clearer and more concise definition on the research questions without diving too much into the technical details.
- Important metrics such as isolation efficiency and tracing efficiency should be clearly defined when it’s first mentioned in the “Results and discussion” section.
- I recommend a dedicated section describing the CNS dataset at the beginning of “Results and discussion” section, followed by a section briefly describing the mathematical models and simulation procedures being used to support the main analysis that produce Fig 2, 4 and 5, with proper reference to the “Data and Method” sections and the “Supplementary Information” for technical details.

Second, the definition of tracing efficiency ϵ_T is very confusing and convoluted. Based on section 4.1.4, it seems like it’s determined by the isolation efficiency and the quarantine policy (Table 1), and thus not an independent factor, while assuming all identified high risk contacts based on a specific policy are being quarantined. In the paper of Ferretti et. al. (1) the tracing efficiency reflects the fraction of contacts being identified comply to quarantine, an very important aspect in real-world implementation of case isolation and contact quarantine. I recommend the authors consider the adoption of a similar definition of tracing efficiency by Ferretti et. al. or introducing a simpler but more intuitive metric relatable to practical execution of contact tracing and quarantine measures.

Finally, I find the authors are missing out an important opportunity provided by the CNS dataset to address an outstanding question on SARS-CoV-2 transmission. In Figure 3, the authors demonstrated the ability of proximity sensors to measure the spatial proximity (signal strength) and duration of exposure of specific social interactions at very high spatiotemporal resolution.

The CNS dataset truly demonstrate the heterogeneities in contact patterns in real-world scenario, with only a small fraction of all contacts occurs within close proximity and prolonged duration of exposure. From an epidemiological point of view, the transmission risk of SARS-CoV-2 is likely highly dependent on the distance and duration of contacts, with closer proximity contacts for a longer duration of exposure posing much higher risk of transmission than other contacts. Such granularity in terms of the resolution of proximity and duration of exposure cannot be obtained through traditional epi-investigation. A proper matching of the tracing policy based on duration and signal strength to the actual spatiotemporal kernel of SARS-CoV-2 transmission risk could potentially significantly reduce the number of individuals required to be quarantined (false positive rate) with minimal sacrifice on the false negative rate. I recommend the authors dive deeper into analysis presented in Figure 5 and systematically analyze scenarios corresponding to different infectiousness dependency of SARS-CoV-2 on the duration and proximity of contacts as well as the optimal policy decision (cut-off on the duration and proximity of contacts measured by the app) that balance the cost (number of individuals need to be quarantined) and effectiveness (infections escape quarantine).

1. L. Ferretti, C. Wymant, M. Kendall, L. Zhao, A. Nurtay, L. Abeler-Dörner, M. Parker, D. Bonsall, C. Fraser, Quantifying SARS-CoV-2 transmission suggests epidemic control with digital contact tracing. *Science* (80-.), eabb6936 (2020).

Reviewer #3:

Remarks to the Author:

Summary:

In "Digital Proximity Tracing in the COVID-19 Pandemic on Empirical Contact Networks" the authors evaluate various digital contact tracing policies and their ability to contain a COVID-19 outbreak. This is not the first manuscript to evaluate digital contact tracing with others clearly demonstrating the difficulty in contact tracing and how digital versions of contact tracing may be more effective. In this vein, the authors build on existing modeling framework to measure the number of newly infected people at time t , $\lambda(t)$, while taking into account the efficacy of isolation policies for infected people and their contacts through digital contact tracing as well as other basic epidemiological parameters. The authors conclude that more restrictive policies are more effective at containing outbreaks, that it is possible to contain an outbreak using less strict policies by considering contacts with longer exposure to infectious individuals and finally that a high level of app participation is crucial to make digital contact tracing effective. These primary conclusions have been reported in other papers (including those cited).

Overall, the modeling and estimation methodology is very difficult to follow, there are several gaps in the procedures for estimating or deriving key parameters and therefore it is very difficult to assess what contributions the authors may have made to existing modeling framework.

Particularly given the abundance of modeling papers – including those focused on contact tracing – currently being published, novelty and public health relevance should be important considerations. As currently written, the novelty of this approach, improvement upon our existing understanding, or additional insight to inform public health practice, are not clear. The lack of clarity in modeling procedure also makes it difficult to assess the degree to which results are artefacts of modeling assumptions. Particular areas that lack information and clarity are addressed point-by-point below. Further, the authors claim to add to existing modeling framework without comparing their results to those obtained by existing modeling frameworks, therefore they have not substantiated this claim. Finally, the implementable policy and public health implications of the author's results are not discussed making the practical utility of their conclusions unclear.

Major Comments:

Parameters that lack clear estimation procedures and information in them modeling framework:

a) EpsilonT: EpsilonT is defined as the ability to trace contacts. The authors use data to estimate the ability to trace contacts based on different thresholds for who would be counted as a contact (different temporal and spatial guidelines). However, this only encompasses a very small portion of tracing contacts. In order for contacts to effectively be traced: 1) proxy/thresholds need to accurately reflect infectious events – i.e. the index case needs to have actually transmitted to the contact, 2) ability to trace contacts – contact them, and 3) the contact to adhere to quarantine orders. All of these factors are included in a single parameter which is varied across simulations. This limits the ability for more realistic and detailed contact tracing scenarios to be investigated. In particular, is it fair to assume that all individuals go into quarantine? What evidence is available (particular from countries such as South Korea) about the actual effectiveness of contact tracing and app-based interventions? A more realistic consideration of these points would help distinguish this work from other, similar work. The terminology in the manuscript – 'the use of real-world data' – is misleading since these data are not following actual contact tracing and infection events, but proximity data. Further, the language describing this parameter (and how it is informed by the data, versus a fixed value used to compare across simulations) is unclear. Two main dependencies of epsilonT are identified as epsilonI and the efficacy of the tracing policy adopted, measured through eT which the authors state is estimated by averaging the fraction of infected people who are infectious and remain unquarantined in numerical simulations run on real temporal networks of contacts. There is no additional information about how these numerical simulations are run nor

about how they estimate eT and no values of eT are given for different modeling scenarios. Therefore, it is impossible to evaluate how eT is estimated, and what eT measures. Since eT is used to compute ϵT it is therefore also impossible to evaluate how ϵT is estimated and what ϵT measures. The evaluation of ϵT is the primary result of their analysis and currently, sufficient detail, clarify, and interpretation are provided to support their conclusions.

b) $\lambda(t)$: The number of newly infected individuals at time (t) which is the main outcome of interest in this study, is derived in the supplement. The authors note that with the derivation of λ described in the supplement, the user no longer needs to consider the asymptotic behavior of the model and can therefore consider real-finite time data. This is useful, however, no information is provided about how this model behaves, whether the values of $\lambda(t)$ obtained from the model are reasonable compared to actual datasets. While it was acceptable earlier in the pandemic to not support results with data when fewer data sets were available, this is no longer true. It is also difficult to assess the potential benefit of this 'novel' modeling approach and how it compares against other published works. Moreover, this novel approach is not addressed in the main body of the paper at all, and is instead relegated to the supplement which is surprising given the importance of these results.

c) Additionally the assumed model relationships between ϵT , ϵI and $\lambda(t)$ are not described clearly in the body of the paper. Without this information in the paper, it is difficult to assess to what degree the results shown are artefacts of model assumptions or to properly understand the general modeling framework. Particularly for models where the outcomes are not fit to actual data, artifacts in models are clearly concerning and should be addressed. The authors do not sufficiently evaluate this issue.

d) The authors use an $R_0 = 2$, which is within the estimated R_0 values, but higher values have been estimated – which should also be considered.

The authors should compare their procedures and results directly to existing studies on digital contact tracing:

a) The authors claim that their modeling framework fills an important gap in existing literature by taking into account features and heterogeneities of real datasets.

1. As described in point a) of the section above about not giving the reader any details about how these numerical simulations of virus spread on real network data are carried out which is important particularly if the authors are claiming this as a major contribution of the work.

2. The authors do not show how their results about contact tracing policies compare with those obtained with previous modeling frameworks. Specifically, the authors need to show what we can learn from their modeling framework is different from what other modeling frameworks have shown.

The authors need to discuss the real-world public health or policy impact of their findings:

a) The authors report in their conclusion section that 80% isolation (ϵI) is required for policies 3,4 and 5 however there is no discussion of how practical 80% isolation may be in different circumstances, nor is there a discussion of how realistic each policy may be under different circumstances.

b) Additionally, policies 2-5 appear to have very similar trajectories and in most scenarios in figure 4. Therefore, it is not clear what practical difference these policies have in terms of outbreak containment which is the primary outcome of interest.

c) The authors also claim that the number of false positive contacts isolated is sensitive to the choice of policy and that therefore fine-tuning chosen policies are important. However, there is no further discussion of what this fine tuning could be and it is not clear how anyone could use the results of the paper to do this.

Minor points:

1. Please replace all instances of she/he with they, them or theirs.

2. Figure 2: the scale of the increase/decrease rate should be made explicit in this figure.

3. Figure 4: the colors are difficult to distinguish and there is no scale accompanying this figure to describe the color differences making it very difficult to infer differences across the three R_0 scenarios.
4. Figure 5: why are false negatives and positives discussed in terms of absolute numbers and not in terms of conditional probabilities i.e. sensitivity specificity?
5. Section A2: table 4: it is not clear why R_0 data corresponding to the model parameter $R_0=2$ is less than R_0 data corresponding to the model parameter $R_0=1.5$.
6. Section C: the authors claim that lengthening the memory of the app contacts to 10 days does not change the results substantially and therefore is not warranted. However, it would be useful to re-do this analysis with $\omega(t)$ that has a thicker tail since the probability mass of this function as described in figure 6 seems to drop off precipitously and therefore it seems that this distribution may be driving the result of section C.
7. What are the shaded areas in Figure 5 and are these just mean simulations?

Response to the Referees' reports

NCOMMS-20-27706

"Digital Proximity Tracing in the COVID-19 Pandemic on Empirical Contact Networks"

We have carefully gone through the reviewers' comments and found their comments to be constructive and insightful. The reviewer's comments have resulted in a completely re-shaped manuscript with the following major changes

- A new Section 2, describing the details of the modeling framework.
- A new analysis of the dependence on the infectiousness on the duration and proximity of the contacts (Section 2.2 and SI A.2), slightly modifying the results shown in the first version of the manuscript.
- A new analysis of digital contact tracing using reduced levels of app adoption (from 20% to 60%) as required by R1.
- A new Section 4.1 on implications and constraints of the policies for digital contact tracing, which discusses socioeconomical constraints and consequences of our policy design.
- A revised and detailed presentation of the fundamental steps of the algorithm in Section 5, including the estimation of the key parameters (Section 5.2).
- A link to an open source version of the code used for the simulations.
- A refined quantification of the size of quarantined population, including a new analysis of the cost and effectiveness of the policies.
- A new analysis modelling the behavior of individuals who reduce their compliance to quarantine after being required to quarantine multiple times (SI C.6).
- A new analysis of the contagion heterogeneity due to social structure (SI D).
- A critical discussion of the comparison with other existing models of digital contact tracing (SI F).

We believe that revised manuscript has been greatly improved by these changes, in particular with respect to the clarity of our arguments, and its usefulness to inform policy designs. We sincerely thank the reviewers for their contribution to the work.

Below, we go over the responses from the three reviewers (*italic text*) point-by-point and provide a response to each specific comment, including a description of the changes implemented.

We hope that the reviewers agree with us that new version of the manuscript is substantially improved and that the manuscript meets the requirements for publication in *Nature Communications*.

With many thanks and best regards,

G. Cencetti, G. Santin, A. Longa, E. Pigani, A. Barrat, C. Cattuto, S. Lehmann, M. Salathé,
B. Lepri

Contents

Response to the report of Referee 1	2
Response to the report of Referee 2	10
Response to the report of Referee 3	13

Response to the report of Referee 1

This is an interesting and original manuscript which addresses important issues related to the current COVID-19 pandemic.

And I wish to thank the authors for writing a so easily readable and understandable manuscript. Their contributions are 3 fold. 1) They extend an existing modeling framework informed by high-resolution contact data. 2) Based on this framework they develop different tracing strategies, and 3) they dig into what costs, in terms of false positives and negatives, each strategy has.

Their work provides considerable improvement of the studies done by Fraser et al. and Ferretti et al. with the authors developing a model that requires no assumption of the functional form of the contagion (growing or decreasing). A further strength of their approach is that parameters ε_I and ε_T no longer need to be assumed as independent.

I think this manuscript would be a good contribution to Nature Communication, however, I only recommend it to be accepted after the below shortcomings are addressed.

We thank the Referee for their positive assessment of the content and contributions of our manuscript, and for the overall appreciation of its solidity and clarity.

The manuscript is very technology centric. The authors present digital contact tracing as an techno-utopia but leave out broader societal discussions of inequality, discrimination, and power. Furthermore, they neglect to discuss the many privacy and ethical issues which have been raised by multiple researchers around contact tracing apps. For examples see below:

- <https://www.nature.com/articles/d41586-020-01578-0>
- https://www.scss.tcd.ie/Doug.Leith/pubs/opentrace_privacy.pdf
- https://www.scss.tcd.ie/Doug.Leith/pubs/contact_tracing_app_traffic.pdf
- [https://www.amnesty.org/\[...\]contact-tracing-apps-danger-for-privacy/](https://www.amnesty.org/[...]contact-tracing-apps-danger-for-privacy/)

I believe the authors have an obligation to mention these issues, because governments and public health agencies might read this manuscript as a validation that contact tracing is effective, when in fact, in most cases it is not a viable solution to the pandemic - as some of the results from this manuscript also show (although this could be discussed more critically).

We agree with the Referee that these issues were not sufficiently discussed in the first version of our manuscript, and we acknowledge their relevance for informing the adoption of digital contact tracing strategies by policy makers. Several of these issues are already taken into account in our modeling framework, but needed to be better highlighted and discussed.

To this end, we have introduced in the “Discussion” a new section entitled “Policies for digital contact tracing: implications and constraints” where we address some important challenges raised by digital contact tracing. First of all, we discuss issues related to the privacy of the individuals: our modeling approach considers indeed a tracing scheme that does not rely on the knowledge of the network of contacts of individuals, but corresponds to a fully decentralized scheme that preserves at best the privacy of users. We then consider the issue of limited app adoption and compliance in the population, which may be due to the limited access to appropriate smartphones for different age and income brackets, but also to people unwilling to adopt the app or the consequent procedures related to the exposure notification step (see also below). The mathematical model includes the effect of limited adoption or compliance, and we have now included additional results for 20% and 40% app adoption. Finally, we comment on the possibility of individuals to isolate (which is encoded within the parameter ε_I), involving both the access to isolated spaces and the economic feasibility of stopping the working activity if necessary, as well as the issues related to equitable testing, since there are countries where this is not being implemented.

– 1) *If we start with the Manuscript title, it is very broad: “Digital Proximity Tracing in the COVID-19 Pandemic on Empirical Contact Networks”. However, the authors only focus on a very specific part of the current COVID-19 pandemic - namely the efficacy of digital contact tracing in the case of re-opening of societies, or places with relatively low infection rates (low R0). The title should reflect these limitations, for instance the title could potentially be: “Digital Proximity Tracing in the COVID-19 Pandemic on Empirical Contact Networks: Controlling re-emerging outbreaks”*

We thank the Referee for the valuable suggestion and we have adopted the proposed title.

– 2) *The authors use very optimistic estimates of adoption rates and smartphone ownership. For digital contact tracing to work, people need access to smartphones and Internet connectivity. However, access to technology is not uniformly distributed across different layers of society. In fact, studies have shown there are wealth-, age-, gender-, ability-, and education-gaps in smartphone ownership. If we look at country specific statistics, in some countries smartphone ownership is below 25%, meaning fewer than 1 in 4 people own a smartphone (see more here → [https://www.pewresearch.org/\[...\]-the-world-but-not-always-equally/](https://www.pewresearch.org/[...]-the-world-but-not-always-equally/)).*

Further, most of the decentralized contact tracing app-frameworks, which the authors describe, rely on BLE technology, which is not available in approx. 2 billion of existing smartphones - predominantly because they are of an older make (read more here → [https://arstechnica.com/\[...\]-contract-tracing-tech/](https://arstechnica.com/[...]-contract-tracing-tech/)). All these issues compound and globally bring down the percentage of individuals who own a smartphone suitable for contact tracing to below 50% of the world population ([https://www.gsma.com/\[...\]GSMA_MobileEconomy2020_Global.pdf](https://www.gsma.com/[...]GSMA_MobileEconomy2020_Global.pdf)).

If we look past smartphone adoption rates and instead focus on whether people are willing to install and use the apps things look even more dire.

In most countries only a small fraction of the populations have installed and use national tracing apps (The most up-to-date estimates of adoption rates, that I’m aware of, can be found here → [https://docs.google.com/spreadsheets/\[...\]](https://docs.google.com/spreadsheets/[...])).

Further, a recent survey for the US shows that 7 out of 10 individuals say they do not even want to install tracing apps ([https://arstechnica.com/\[...\]data-shows/](https://arstechnica.com/[...]data-shows/)).

Even in a country like Germany, with relatively high trust in the government, the national tracing app has only been installed 13 million times (not to be confused with it being installed by 13

million people), compared to the total population of 83 million, this amounts to a very low 16% adoption rate.

All these factors compound to lower the adoption rate of contact tracing apps. While the authors consider different adoption rates and acknowledge that “high level of app adoption is crucial to make digital contact tracing an effective measure”, their most conservative estimate of 60% is very far from reality.

To have an open and unbiased discussion around the usefulness of contact tracing the authors need to include adoption rates of 20% and 40% in their study. Fig 4 would be a good place to depict the effect of low-adoption rates. For instance, by adding two additional columns to the figure.

We agree with the concerns raised by the Referee, and thus we have reported and discussed additional results. The main text now shows three different scenarios of compliance: 20%, 40% and 60% of app adoption (see novel version of Fig. 4), while the unrealistic results with 80% and 100% were removed.

– 3) For contact tracing apps to work there needs to be access to equitable testing. In the manuscript the authors indirectly assume this, but never explicitly mention that without widespread and equitable testing contact tracing frameworks won’t work.

This ties in with one of the key parameters of the model, $s(\tau)$ the probability for an infected individual to be recognized as infected within a period of time τ , via some form of testing regiment. As a value of τ the authors use 2 days, which in a “normal” setting seems like a reasonable choice. However, experiences from different countries show that this is not the case. In the US, for instance recently released statistics show that the average turnaround time for a test is 7 days ([https://www.cnbc.com/\[...\]-the-nation-quest-diagnostics-says.html](https://www.cnbc.com/[...]-the-nation-quest-diagnostics-says.html)).

How robust are the results to settings in which cases are reported with a delay of more than 2 days?

We thank the Referee for this valuable suggestion. In the revised version of the paper, we have discussed the results of additional simulations with a higher delay and observed that even a delay of 3 days reveals a much worse scenario, making it very difficult to contain the epidemic. This is reported in the Supplementary Information, Section C.2.

– 4) The work builds on the assumption that individuals:

1) are rational agents who will optimize common good through self-quarantine once notified of exposure

2) have access to a safety net in order to properly quarantine

While I was glad to see that the authors try to model how many people can self-isolate through the parameter ϵ_1 . This parameter will change (and probably decay over time) if individuals are notified multiple times. Basically people will reach a level of fatigue and stop following the notifications from the app, once they have been notified for the n 'th time.

Only the most well-off individuals and people living in societies with social and labour safety nets will be able to properly follow the recommendations from the apps. For instance, we know that people working in the most vulnerable jobs (cashiers, bus-drivers, teachers) will be notified more frequently than the average person, but they will not necessarily have the means to self-quarantine.

While a comprehensive study of this is out of the scope for this manuscript, I would like to see statistics for how many times individuals are asked to self-isolate. E.g. run the simulations over 50 days, with a self-quarantine period of 14 days, what will be the frequency distribution of exposures? Put differently, out of the 706 participants from the Copenhagen Network Study how many of the participants will receive the notification once, twice, thrice, etc.?

We thank the Referee for this useful suggestion. We have inserted a further analysis (i.e., making use of an extended version of the CNS dataset) counting the number of times that an individual is asked to quarantine, and taking into account the fact that compliance to quarantine will decrease if an individual is wrongly notified multiple times.

To effectively account for a reduction of the compliance over time, we have modeled a discount factor that reduces the compliance of an individual over successive quarantines' requests in the numerical simulation. This has an impact on the definition of ε_T , while the theoretical model is not affected. See section C.6 of Supplementary Information.

– 5) *Comparing tracing strategies to full lockdowns.*

In Figure 3 the authors show, that depending on the tracing strategy, thousands to tens of thousands of contacts could be flagged as contagion events. Since the study population is relatively small (706 individuals for the Copenhagen Network Study, and different for the other datasets), I would like to see how large a fraction of the full population each strategy would require to be in quarantine over time. This will enable the reader to compare digital tracing to a full lockdown.

To a certain degree the authors try to show this in Fig 5, however, they never show how large a fraction of the body of participants that would mean.

Judging from the table in Fig 5 the results seem to suggest that policies 4 and 5 require around 40 – 60% of the study participants (population) to be in quarantine - which is not very different from a full lockdown. The only difference is that this “hybrid lockdown” with people going in and out of quarantine would be a logistical nightmare.

Can you please do a plot, comparing the 5 strategies, with time on the x-axis (0-50 days), and the cumulative proportion of population that has been in lockdown on the y-axis?

Following the Referee' suggestion we have restructured the figures presenting the time evolution of the number of quarantined individuals. Namely, in the main text (see e.g. Figure 5) we report the percentage of false negatives and false positive over time (i.e., the fraction over the entire population) instead of the absolute numbers. This visualization gives a better glimpse at the relative size of the set of quarantined people. Moreover, we added a figure that reports the number of total quarantined individual, meaning the number of unique individuals that are quarantined in the overall simulation time (which is compared to the number of false negatives). Finally, we included also a table with the total percentage of quarantined individuals with respect to the entire population, and the percentage of infected individuals over the number of quarantined individuals. The numbers in this table are particularly interesting because they show that at most 9% of the entire population is quarantined (in the case of the most restrictive policy), making the scenario very different from a total lockdown.

However, we would like to remark that focusing on the specific result of a numerical simulation could lead to erroneous or inaccurate conclusions and it is more appropriate to use the simulations as a proxy to compare different settings.

Indeed, the main result that we extrapolate from numerical simulations on the datasets is the tracing ability ε_T , which we then insert into the mathematical model, thus obtaining a general formulation that is itself not strictly bounded by any specific dataset. The other results obtained by the raw simulations should be considered partial, as they are performed on a specific contact network involving only university students, i.e., they correspond only to a partial knowledge of the whole society contact network. For example, to quarantine

40 – 60% of a population of students in the case of an outbreak in this population is not the same as quarantining 40 – 60% of the entire population within a country.

– 6) *Repeatability and context of contacts. Students in the Copenhagen Network Study will most likely interact with the same individuals across time, due to heterogeneities created by class structures, studylines, seniority of students, etc.*
As such individuals will most likely be exposed to contagion events within these structures.
How effective are the five strategies at identifying contagion events across these heterogeneities?
I.e. how large a fraction of the events depicted in Fig 3 can be attributed to within-group vs between-group?

As mentioned in the previous point, we wish to remark that the model’s conclusions are not bounded to the specific data set (see e.g. the discussion in the new Section 4.2 “Digital contact tracing: insights and limitations”).

The Referee’s point is nevertheless very interesting and we have investigated it. Separating the individuals in groups, defined by applying a well-known algorithm for community detection to the network, we separated the contagion events that happen via intra- and inter-group interactions. We then reported the evolution of the spread in each group, observing that some groups tend to become more rapidly infected, while some others prove instead more resistant to contagion. See Supplementary Information, section D.

– 7) *There is no discussion around masks, which I find very disturbing.*
In their results the authors show that digital contact tracing works when, and only when, the following conditions are met:
1) high smartphone ownership and app adoption, preferably higher than 60% of population - which no country has achieved so far
2) randomized mass testing schemes, to enable that many asymptomatic carriers are discovered and quarantined - very few places have the resources to do this
3) a social safety net where people have the resources to self-isolate when notified of exposure - very rare outside of Europe (even in Europe, particularly for the most vulnerable populations)
4) high tracing and isolation efficiencies (ϵ_T and ϵ_I)
5) Limited spread of COVID, i.e. Low reproduction numbers (low R_0 values)
Based on these assumptions the authors conclude that digital contact tracing can be used as another non-pharmaceutical intervention to reduce the spread of COVID.
I think this is a fallacious conclusion, because advocating for deploying promising but untested technologies without mentioning masks is deceiving.
Masks work to stop the spread of COVID no matter whether people have smartphones or not, or whether R_0 is small enough, or whether there is enough mass testing.
The authors need to put the “effectiveness” of digital contact tracing into perspective to the efficacy of low-tech solutions such as masks. And they also need to be more critical in discussing their results.
For more info about low-tech solutions that authors can read more here →
[https://www.thelancet.com/\[...\]PIIS0140-6736\(20\)31142-9/fulltext](https://www.thelancet.com/[...]PIIS0140-6736(20)31142-9/fulltext).

We thank the Referee for pointing out this important issue, as we should have been more explicit on discussing this matter.

In general terms, the paper goal is not to promote digital contact tracing as the unique solution for pandemic containment, but rather instead to analyze if, when, and how much

digital contact tracing is effective depending on complementary conditions associated to other measures. In other words, we want to understand when digital contact tracing is able to complement these measures to contain the pandemic.

In particular, we have considered several levels of reduced transmissibility by using lower values of R_0 , and these reduced values are due to generic mitigation measures (i.e., not due to contact tracing) such as limitations of gatherings, mask wearing, etc. Our results actually clearly show that even stringent digital contact tracing policies are not enough if R_0 is large, meaning that contact tracing can only act as an additional measure in a situation where several “low-tech” measures are already in place in order to decrease R_0 . Among these measures, it is clear that masks play a very important role, and we now make this point explicit.

– 8) *Technical Issues.*

The assumption of the authors policies builds upon a dataset where all users have identical devices (smartphones in the case of the Copenhagen Network Study or RFID tags in the case of the supplementary datasets), but this is not true in real-world settings. How well do the authors think this will work for a setting where individuals have different smartphone brands?

The issue of the non-uniformity of Bluetooth signal and sensors across different devices is certainly a serious one, and we are fully aware of it, as are the developers of the apps in the various countries: an important empirical work of calibration in realistic settings has been performed in a number of countries before deciding which RSSI thresholds to use.

Note that, in fact, this problem appears even with identical devices (that may have slightly different sensors) and different studies have highlighted the difficulty of establishing a distance measure out of Bluetooth interactions, due e.g. to the interaction with other waves and obstacles that the signal can meet along its path. This is one of the reason we consider here only signal strength levels, without attempting to convert them to physical distances: actual apps do exactly this, considering as “at-risk” contacts with RSSI values above a given threshold for a certain time duration.

We have discussed this in the revised version of the paper in Section 2.3 (Design of appropriate policies).

– 9) *The proposed model and corresponding code provides a great testbed to test additional scenarios and contact networks. Will the authors open source the model code?*

The code is indeed public: https://github.com/DigitalContactTracing/covid_code, and we have added an explicit pointer to this repository in the revised version of the paper.

Minor issues:

1. *Using the term “social distancing” sends the wrong message. While “social distancing” was initially widely used the conversation has shifted towards “physical distancing”.*

This is reflected in the WHO’s guidelines ([https://www.who.int\[...\]-distancing](https://www.who.int[...]-distancing)) which also say that while people need to physically distance they should stay socially connected. Please update your manuscript to reflect this.

We thank the Referee for pointing out this issue. We use the term “physical distancing” in the revised version of the manuscript.

2. You are inconsistent in how you write lockdown and use both “lock-down” and “lockdown” in the text. I suggest you change lock-down(s) → lockdown(s) on l.17, l.38, l.46, l.48, l.131, l.496, and l.953

We have modified it according to the Referee’s suggestion.

3. l.38 prove → proven to be

We have fixed it in the revised version of the manuscript.

4. l.152 covid-19 → COVID-19 (to be consistent in how you refer to the virus in other places in the MS and also to align with the official name of the disease) Same on l.170, l.242, l.339

We have fixed it in the revised version of the manuscript.

5. l.160 his/her → their

We have fixed it in the revised version of the manuscript.

6. l.161 s/he → they

We have fixed it in the revised version of the manuscript.

7. You never reference Fig 1 in the manuscript

Thank for pointing out this issue. We have fixed it in the revised version of the manuscript.

8. l.197 “below the thresholds” → “below an RSSI threshold”

We have fixed it in the revised version of the manuscript.

9. I’m nitpicking here, but it might make sense to switch the ordering of figures 2 and 3, to reflect the order you refer to them in the text.

We agree with the Referee and we have switched the ordering of Figures 2 and 3.

10. On l.449 you say the the Copenhagen Network Dataset contains contact information for 1 month, however, on l.225, you model the infection up until $T = 50$ days. How? If this is because you do not use contact networks for the idealized scenario it would be good to indicate this directly in the text.

Yes, we are using the model and not the dataset. We have clarified this point in the revised version of the manuscript.

11. l.255, you are referring to a Table in Figure 4, but Fig 4 has no table.

We have fixed it in the revised version of the manuscript.

12. Fig 4, could you please add the respective colorbars for each of the subplots, or just one for all of the plots?

We thank the Referee for pointing out this issue. We have added a common colorbar for all the subplots of Figure 4.

13. Results of simulations in Fig 4. I am assuming you have done multiple simulations (since they tend to be stochastic) and averaged over them, could you please indicate the errorbars for the individual points in the Figure.

Thank for pointing out this issue. We have added errorbars in Figure 4.

14. Fig 5, the false negative and false positive plots, what are the y-axes showing? Is this infection events (edges) or infected individuals (nodes)?

They represent infected individuals (nodes). We have made this point clear in the revised version of the manuscript.

15. L.321-323. The sentence "For all other policies instead the curves of false negatives all reach similar levels" says that policies 2-5 all have similar levels of false negatives. However, the next sentence then says that "The curve drops to zero rapidly however only for the stricter policies, while for Policy 2 it remains at a higher level, ..." meaning that Policy 2 is different from Policies 3-5. These two sentences are confusing and inconsistent, you cannot argue that Policies 2-5 are the same, and then say that Policy 2 is different from Policies 3-5.

We thank the Referee for spotting these confusing comments, we have revised the text accordingly.

16. l.464, the weighted network you are constructing is not necessarily represented by a distance weighted connections. There are lots of issues with interpreting RSSI as physical distance in complex environments (lots of concrete, metal, etc), see more here → <https://arxiv.org/pdf/2006.08543.pdf>. Instead your edge weights stand for "temporal and signal strength-weighted connections".

We agree with the Referee, as discussed above. Our measures are based on signal strength and not on physical distances. We have made clear this in the revised version of the manuscript, using the formulation suggested by the Referee.

17. On l.467 you are saying that simulations start from an initial number of infected people, Y_i , however, you never mention how Y_i is chosen/sampled

Y_i is one of the parameters of the model. In the simulations we always consider $Y_i = 1$. We have clarified this point in the revised version of the manuscript.

Response to the report of Referee 2

In the manuscript titled “Digital Proximity Tracing in the COVID-19 Pandemic on Empirical Contact Networks”, the authors utilize mathematical models to explore the feasibility to achieve SARSCoV-2 control based on digital contact tracing app, utilizing proximity sensors commonly available in smartphones. The topic of the manuscript is interesting and relevant to the current situations of the SARS-CoV-2 pandemic, as many countries and public health agencies have developed and deployed digital tracing apps to assist epidemiological investigation and control. The key contribution of the study is the utilization of interaction data collected by the “Copenhagen Network Study” prior to the COVID-19 pandemic, which is also collected through smartphones. The dataset provides a very unique perspective on the spatiotemporal granularities of the interaction data that a commonly available smartphone today is capable of capturing and cannot be obtained through traditional epidemiological investigation. The manuscript has the potential to make an important advancement in this particular area of research. However, there are a few major issues that need to be addressed before it can be considered to be published at Nature communication.

We thank the Referee for their careful reading and for appreciating our contribution to this area of research. We have significantly revised the manuscript trying to improve it according to the Referee’s suggestions.

First, improvement in writing is recommended for the authors before the manuscript could be considered for publication. It’s understandable that during the pandemic era the authors are challenged to put together a rather complex study given limited time. However, the manuscript at current stage is quite difficult to follow overall. Specifically:

- *The “Introduction” section could be shortened (3 pages currently) with a clearer and more concise definition on the research questions without diving too much into the technical details.*
- *Important metrics such as isolation efficiency and tracing efficiency should be clearly defined when it’s first mentioned in the “Results and discussion” section.*
- *I recommend a dedicated section describing the CNS dataset at the beginning of “Results and discussion” section, followed by a section briefly describing the mathematical models and simulation procedures being used to support the main analysis that produce Fig 2, 4 and 5, with proper reference to the “Data and Method” sections and the “Supplementary Information” for technical details.*

We thank the Referee for these valuable suggestions. We have shortened and significantly revised the Introduction Section. Moreover, we have reorganized the text moving part of the material from the Supplementary Information to the main paper. In particular, we have introduced the mathematical model showing the equations already in the main text (new Section 2), relegating to the Supplementary Information only the derivation and the mathematical details. We now give a clear definition of the basic quantities introduced, like ε_T and ε_I (new Section 2.1). We also provide a detailed description of the datasets and of the numerical simulation.

Second, the definition of tracing efficiency ε_T is very confusing and convoluted. Based on section 4.1.4, it seems like it’s determined by the isolation efficiency and the quarantine policy (Table 1), and thus not an independent factor, while assuming all identified high risk contacts based on a

specific policy are being quarantined. In the paper of Ferretti et. al. (1) the tracing efficiency reflects the fraction of contacts being identified comply to quarantine, an very important aspect in real-world implementation of case isolation and contact quarantine. I recommend the authors consider the adoption of a similar definition of tracing efficiency by Ferretti et. al. or introducing a simpler but more intuitive metric relatable to practical execution of contact tracing and quarantine measures.

1. L. Ferretti, C. Wymant, M. Kendall, L. Zhao, A. Nurtay, L. Abeler-Dörner, M. Parker, D. Bonsall, C. Fraser, *Quantifying SARS-CoV-2 transmission suggests epidemic control with digital contact tracing*. *Science* (80-.), eabb6936 (2020).

This definition is indeed central in our work. In particular, we considered the compliance as encoding the compliance to all parts of the contact tracing and quarantine procedure. In other words, if someone installs the app but then does not quarantine if notified she/he should be counted among the non-compliant individuals since the effect would be the same than that of not adopting the app. The non-compliance (or impossibility) to quarantine is therefore already considered when choosing the percentage of app adoption. Similarly, the non-compliance to isolate, if recognized as infected, is encoded into the isolation efficiency ε_I . We have made this clearer in the revised version of the manuscript.

In addition, given the relevance of this distinction to correctly define the tracing procedure, and after a request of Referee 1, we have performed a numerical simulation (see Supplementary Information, section C.6) where we inserted a distinction between non-adopters of the app and non-compliance to quarantine. Following the suggestion of Referee 1, we have moreover taken into account that compliance to quarantine decreases if the same person is wrongly notified multiple times.

Finally, I find the authors are missing out an important opportunity provided by the CNS dataset to address an outstanding question on SARS-CoV-2 transmission. In Figure 3, the authors demonstrated the ability of proximity sensors to measure the spatial proximity (signal strength) and duration of exposure of specific social interactions at very high spatiotemporal resolution. The CNS dataset truly demonstrate the heterogeneities in contact patterns in real-world scenario, with only a small fraction of all contacts occurs within close proximity and prolonged duration of exposure. From an epidemiological point of view, the transmission risk of SARS-CoV-2 is likely highly dependent on the distance and duration of contacts, with closer proximity contacts for a longer duration of exposure posing much higher risk of transmission than other contacts. Such granularity in terms of the resolution of proximity and duration of exposure cannot be obtained through traditional epi-investigation. A proper matching of the tracing policy based on duration and signal strength to the actual spatiotemporal kernel of SARS-CoV-2 transmission risk could potentially significantly reduce the number of individuals required to be quarantined (false positive rate) with minimal sacrifice on the false negative rate. I recommend the authors dive deeper into analysis presented in Figure 5 and systematically analyze scenarios corresponding to different infectiousness dependency of SARS-CoV-2 on the duration and proximity of contacts as well as the optimal policy decision (cut-off on the duration and proximity of contacts measured by the app) that balance the cost (number of individuals need to be quarantined) and effectiveness (infections escape quarantine).

We thank the Referee for raising this interesting point. It is true that in the original version of the manuscript we assumed a specific shape of the infection curve and we only considered one possibility for its dependency on proximity and duration of contacts.

In the revised version of the paper, we have explored different scenarios in order to provide a deeper investigation of the spatio-temporal patterns of contagion and consequent policies to be adopted. More in details, we have investigated the analytical dependence of the infection curve on some physical constraints on both proximity and duration. Furthermore, we have performed numerical simulations for different scenarios, each one corresponding to different physical constraints. We have included and discussed the results of these additional simulations in the new version of the paper (section A.2 of Supplementary Information). We have also provided a clearer graphical representation of cost and effectiveness for each policy (see for instance Fig. 5 and similar figures in the Supplementary Information).

Response to the report of Referee 3

Summary:

In “Digital Proximity Tracing in the COVID-19 Pandemic on Empirical Contact Networks” the authors evaluate various digital contact tracing policies and their ability to contain a COVID-19 outbreak. This is not the first manuscript to evaluate digital contact tracing with others clearly demonstrating the difficulty in contact tracing and how digital versions of contact tracing may be more effective. In this vein, the authors build on existing modeling framework to measure the number of newly infected people at time t , $\lambda(t)$, while taking into account the efficacy of isolation policies for infected people and their contacts through digital contact tracing as well as other basic epidemiological parameters. The authors conclude that more restrictive policies are more effective at containing outbreaks, that it is possible to contain an outbreak using less strict policies by considering contacts with longer exposure to infectious individuals and finally that a high level of app participation is crucial to make digital contact tracing effective. These primary conclusions have been reported in other papers (including those cited).

We thank the Referee for these comments and we agree that some of the results we describe in our manuscript are in line with those presented in other papers and pre-prints. As also appreciated by the two other Referees, we believe that this work addresses a number of novel aspects that deserve to be investigated by the researchers working to build a better understanding of digital contact tracing solutions as well as by policy makers interested in adopting this approach in conjunction with traditional contact tracing.

In particular, the main distinctive characteristics of our work are as follows: (i) a general mathematical model that allows to evaluate the evolution of an epidemic in presence of isolation and tracing for any shape of the epidemic growth (which can therefore be obtained numerically or by theoretical assumptions); (ii) the evaluation of tracing efficiency by means of a numerical simulation on real contact data, and no more on an arbitrary parameter of the model; (iii) the dependence on real duration of exposure time and on real physical proximity of contacts; (iv) the corresponding design of appropriate policies.

Moreover, as suggested by the second Referee we fully exploited the opportunities provided by the CNS data set and we included in the paper an analysis of the dependence of infectiousness on the duration and proximity of the contacts (new Section 2.2).

Overall, the modeling and estimation methodology is very difficult to follow, there are several gaps in the procedures for estimating or deriving key parameters and therefore it is very difficult to assess what contributions the authors may have made to existing modeling framework. Particularly given the abundance of modeling papers – including those focused on contact tracing – currently being published, novelty and public health relevance should be important considerations. As currently written, the novelty of this approach, improvement upon our existing understanding, or additional insight to inform public health practice, are not clear. The lack of clarity in modeling procedure also makes it difficult to assess the degree to which results are artefacts of modeling assumptions. Particular areas that lack information and clarity are addressed point-by-point below. Further, the authors claim to add to existing modeling framework without comparing their results to those obtained by existing modeling frameworks, therefore they have not substantiated this claim. Finally, the implementable policy and public health implications of the author’s results are not discussed making the practical utility of their conclusions unclear.

In a context of an abundance of modeling papers, it is right that highlighting the novelty of our study and clearly stressing the implications of our findings for practical policy design,

are of utmost importance.

We strongly believe that our work contains such novelty and relevant implications and, as also suggested by the other Referees, we have reorganized the main text in order to make the presentation clearer. In particular, we have moved part of the material from the Supplementary Information to the main text, thus introducing the mathematical model and the equations already in the main text. Instead, we have relegated to the Supplementary Information only the derivation and the mathematical details. In the revised version of our manuscript, we have also provided a clearer definition of the basic quantities introduced, such as ε_T and ε_I , as well as a detailed description of the datasets and of the numerical simulation (see sections 2 and 5).

Moreover, we clarified that the works of Fraser et al. [RF2] and Ferretti et al. [RF1] represent the baseline of our model. We have however built on top of this model a general architecture that allows us to go beyond the original analyses and open a wider scenario for policy evaluation. The original framework proposed by Fraser et al. and Ferretti et al. is indeed not general enough to be used in a re-emerging scenario, being governed by an exponential evolution of infections. Moreover, by discretizing the equations we adapted it to numerical simulations. This in turn allows to use real contact networks and the corresponding metadata. We propose an epidemic model based on duration and proximity of contacts, and as a consequence we are able to devise policies based on duration and proximity thresholds. Those policies can be evaluated based on efficacy in containing the spread and social cost (in terms of false positive in quarantines). All these analyses and results could not be obtained with the original framework proposed by Fraser et al. [RF2] and Ferretti et al. [RF1], and for this reason it is not possible to depict an actual comparison between these two models. Nevertheless, we added Section F to the Supplementary Information, where we give a detailed overview of the state of the art on contact tracing models, highlighting that none of the published papers (up to our knowledge) deal with the aspects that we analyze. Even those using real-world data do not perform similar analyses and in particular there is no equivalent work designing and evaluating digital contact tracing policies based on duration and proximity of contacts.

Finally, we modified the discussion section and our conclusions are now clearer and suggest specific strategies for policy makers.

Parameters that lack clear estimation procedures and information in their modeling framework:
a) ε_T : ε_T is defined as the ability to trace contacts. The authors use data to estimate the ability to trace contacts based on different thresholds for who would be counted as a contact (different temporal and spatial guidelines). However, this only encompasses a very small portion of tracing contacts. In order for contacts to effectively be traced: 1) proxy/thresholds need to accurately reflect infectious events – i.e. the index case needs to have actually transmitted to the contact, 2) ability to trace contacts – contact them, and 3) the contact to adhere to quarantine orders. All of these factors are included in a single parameter which is varied across simulations. This limits the ability for more realistic and detailed contact tracing scenarios to be investigated. In particular, is it fair to assume that all individuals go into quarantine? What evidence is available (particular from countries such as South Korea) about the actual effectiveness of contact tracing and app-based interventions? A more realistic consideration of these points would help distinguish this work from other, similar work. The terminology in the manuscript – ‘the use of real-world data’ – is misleading since these data are not following actual contact tracing and infection events, but proximity data. Further, the language describing this parameter (and how it is informed by the

data, versus a fixed value used to compare across simulations) is unclear. Two main dependencies of ε_T are identified as ε_I and the efficacy of the tracing policy adopted, measured through e_T which the authors state is estimated by averaging the fraction of infected people who are infectious and remain unquarantined in numerical simulations run on real temporal networks of contacts. There is no additional information about how these numerical simulations are run nor about how they estimate e_T and no values of e_T are given for different modeling scenarios. Therefore, it is impossible to evaluate how e_T is estimated, and what e_T measures. Since e_T is used to compute ε_T it is therefore also impossible to evaluate how ε_T is estimated and what ε_T measures. The evaluation of ε_T is the primary result of their analysis and currently, sufficient detail, clarify, and interpretation are provided to support their conclusions.

Regarding the Referee sentence “proxy / thresholds need to accurately reflect infectious events – i.e. the index case needs to have actually transmitted to the contact”, we would like to point out that in real world we do not know a priori which contacts have been contagious and the challenge is to guess which policies to follow in order to better identify the possibly infected individuals, overcoming this lack of information. This is exactly what we want to reconstruct with our analysis: we assume that infection follows some rules (also based on distance and proximity) and simulate the epidemics, then we suppose we do not know which individuals are infected and measure how well different tracing policies are able to optimize the finding of the actually infected contacts (also taking into account that we cannot quarantine all the contacts, otherwise it would be a lockdown). We have clarified this point in the revised version of the paper (new Section 2.2 and Section 5.1)

Moreover, Referee 3 criticizes the fact that we assume that everyone can be contacted if identified as possible infected, but actually we are assuming that only those who make use of the app can be notified, while the others are never reached or traced. Moreover, in general we consider that compliance to self-quarantine once notified is included in the “adoption” parameter: we implicitly assume that when someone download the app s/he agrees to the rules and will quarantine if notified. In the revised version of the manuscript, also following the suggestions of the other Referees, we have inserted in the Supplementary Information (section C.6) a numerical simulation with a distinction between non-adopters of the app and non-compliants to quarantine, considering that compliance to quarantine decreases if the same person is wrongly notified multiple times.

Finally, we agree with the Referee that the definitions of ε_T and e_T were not clear enough and we have improved this point in the revised version of the manuscript (new Section 2.1 and Section 5.2).

Finally regarding the use of real data, we make clear in the manuscript that the data we use concern contact data in a real-world population, and not data of actual contagion events. Indeed, as in other data-driven models, we consider the contact data as the temporal network on which a contagion process can occur. It corresponds exactly to the type of contacts that could be measured by an app and thus would be used in a real contact tracing procedure. Thus, we stand by our claim that we use “real-world data” that contains all the rich features, heterogeneities and structures that characterize contacts between individuals and are not included in homogeneous mixing approaches.

Besides, the reason why we did not use real contagion contact data is that, to the best of our knowledge, such data are simply not available. Indeed such data have never been used in published papers.

b) $\lambda(t)$: The number of newly infected individuals at time (t) which is the main outcome of interest in this study, is derived in the supplement. The authors note that with the derivation of λ described in the supplement, the user no longer needs to consider the asymptotic behavior of the model and can therefore consider real-finite time data. This is useful, however, no information is provided about how this model behaves, whether the values of $\lambda(t)$ obtained from the model are reasonable compared to actual datasets. While it was acceptable earlier in the pandemic to not support results with data when fewer data sets were available, this is no longer true. It is also difficult to assess the potential benefit of this ‘novel’ modeling approach and how it compares against other published works. Moreover, this novel approach is not addressed in the main body of the paper at all, and is instead relegated to the supplement which is surprising given the importance of these results.

We thank the Referee for their suggestion, and we indeed recognize the importance of this step in our modelling procedure. We have moved part of the derivation and explanation of the model to the main text.

However, we would like to clarify that the novelty of our approach does not concern the theoretical method of derivation of $\lambda(t)$, but rather the coupling of the model with a network obtained from real contact data. Indeed, the equations that compute λ are well-established and widely used and have been published in high-level peer reviewed journals [RF3, RF2, RF1]. At this level we adopt them exactly as they are, and in this respect we believe that no comparison with other datasets is required to validate them.

Instead we have introduced a new discretization of the model into a finite set of linear equations that can be solved numerically. This step allows us to simulate the model with more freedom on the choice of the parameters (e.g., pre-asymptotic behavior). This second step is instrumental for the real novelty of the paper. Indeed, as noted by the other Referees, the main contribution of the paper is the coupling of the model for λ with a realistic study of the key parameters obtained by the simulation on real proximity data. Having a more flexible discretization of the model allows us to be more adaptive to the outcomes of these simulations. We have made this clear in the revised version of the manuscript.

c) Additionally the assumed model relationships between ε_T , ε_I and $\lambda(t)$ are not described clearly in the body of the paper. Without this information in the paper, it is difficult to assess to what degree the results shown are artefacts of model assumptions or to properly understand the general modeling framework. Particularly for models where the outcomes are not fit to actual data, artifacts in models are clearly concerning and should be addressed. The authors do not sufficiently evaluate this issue.

We have significantly reorganized the text in order to provide a clearer and more accurate definition of all the model quantities. In particular, the new Section 2 is entirely devoted to the detailed explanation of the model.

d) The authors use an $R_0 = 2$, which is within the estimated R_0 values, but higher values have been estimated – which should also be considered.

We understand (and already acknowledge in the paper) that the choice of R_0 is probably one of the few crucial ingredients that mostly determines the outcome of our analysis. In this view, we have decided to concentrate on scenarios of post-pandemic contact tracing,

where the infectiousness is already weakened by multiple prevention procedures, such as mask wearing, physical distancing, avoiding large gatherings, etc., that we now discuss in more details. Thus, we restricted our analysis to values of R_0 that are compatible with this choice: $R_0 = 1.2, 1.5, 2$. Following the suggestion of Referee 1, we have also changed the title of the paper to “Digital Proximity Tracing in the COVID-19 Pandemic on Empirical Contact Networks: Controlling re-emerging outbreaks”.

The authors should compare their procedures and results directly to existing studies on digital contact tracing:

a) The authors claim that their modeling framework fills an important gap in existing literature by taking into account features and heterogeneities of real datasets.

1. As described in point a) of the section above about not giving the reader any details about how these numerical simulations of virus spread on real network data are carried out which is important particularly if the authors are claiming this as a major contribution of the work.

2. The authors do not show how their results about contact tracing policies compare with those obtained with previous modeling frameworks. Specifically, the authors need to show what we can learn from their modeling framework is different from what other modeling frameworks have shown.

As previously highlighted, a distinctive characteristic of our work is the evaluation of tracing efficiency on real contact data captured by Bluetooth sensors, instead of treating it as an arbitrary parameter of the model. In particular, our work focuses on investigating how much the efficiency of digital contact tracing is influenced by the definition of different thresholds on the duration of exposure time and on the physical distance of detected contacts. To the best of our knowledge, there are no previous published works focused on an evaluation of efficiency of digital contact tracing strategies with real data collected using signals similar to the ones currently used by the actual contact tracing apps deployed in several countries. Thus, a generic comparison of our modeling framework with previous modeling approaches is out of the scope of this investigation. However, following the Referee’ suggestion, we have introduced Section F in the Supplementary Information to discuss the state of the art in tracing models, and we have substantiated the claim that our model uses the existing framework proposed by Ferretti et al. as a baseline but the main results stem from the modifications that we add on top of that model. We indeed bring the model to a further level where it can be coupled with real-data to obtain a realistic implementation of the epidemic spread, thus allowing appropriate policy design and evaluation.

For what concerns the numerical simulations, the method is now explicitly described in section 5.

The authors need to discuss the real-world public health or policy impact of their findings:

a) The authors report in their conclusion section that 80% isolation (ϵ_1) is required for policies 3,4 and 5 however there is no discussion of how practical 80% isolation may be in different circumstances, nor is there a discussion of how realistic each policy may be under different circumstances.

In the revised version of the manuscript, we have added a section entitled “Policies for digital contact tracing: implications and constraints” where this topic is discussed. Clearly each country has a different level of capacity to isolate individuals and the applicability of the policies that we hypothesize changes from situation to situation. This uncertainty is one of

the reasons that motivates our interest in the study and comparison of multiple scenarios (e.g., several levels of ϵ_I), instead of prescribing a fixed setting: Reporting such (possibly extreme) values of the parameters is in our view a strength, and not a weakness of our work.

b) Additionally, policies 2-5 appear to have very similar trajectories and in most scenarios in figure 4. Therefore, it is not clear what practical difference these policies have in terms of outbreak containment which is the primary outcome of interest.

It is indeed the case that from the point of view of Figure 4 (epidemic containment) these policies are very similar. They nevertheless significantly differ in the number of false positives that they produce (Figure 5).

This is indeed one of the key findings of our work: policies that may look similarly effective may imply different social costs. Thus, evaluating and choosing a policy requires to consider a full spectrum of aspects. One possibility is to evaluate efficiency and cost from graphical representations like that in Figure 5 (c).

c) The authors also claim that the number of false positive contacts isolated is sensitive to the choice of policy and that therefore fine-tuning chosen policies are important. However, there is no further discussion of what this fine tuning could be and it is not clear how anyone could use the results of the paper to do this.

This point is related to the previous one, and the same response partially applies. To make this aspect fully clear, we deeply modified Section 4 “Discussion” and we make it clear how different aspects of the model should be linked to socioeconomic considerations. In particular, as highlighted in this new section, we wish to remark that this flexibility is a strength of our modelling effort, in that it allows to tune policies on the ground of quantitative results in terms of effectiveness (i.e., epidemic containment) and cost (i.e., number of quarantines).

Minor points:

1. Please replace all instances of she/he with they, them or theirs.

We have fixed them in the revised version of the paper.

2. Figure 2: the scale of the increase/decrease rate should be made explicit in this figure.

We are not sure we understand this point, in old Figure 2 (now Figure 1) the three panels are accompanied with a colorbar indicating increase and decrease rate.

3. Figure 4: the colors are difficult to distinguish and there is no scale accompanying this figure to describe the color differences making it very difficult to infer differences across the three R0 scenarios.

We thank the Referee for pointing out this issue. We have added a color scale.

4. Figure 5: why are false negatives and positives discussed in terms of absolute numbers and not in terms of conditional probabilities i.e. sensitivity specificity?

We thank the Referee for raising this point. We modified the relevant figures (mainly Figure 5 in the main text) to report percentages of quarantined individuals.

5. Section A2: table 4: it is not clear why R_0 data corresponding to the model parameter $R_0=2$ is less than R_0 data corresponding to the model parameter $R_0=1.5$.

We have made this point clear in the new version of the manuscript. Now the different values of R_0 are obtained by tuning a multiplying factor r_{R_0} , as reported in Table 6 of the Supplementary Information.

6. Section C: the authors claim that lengthening the memory of the app contacts to 10 days does not change the results substantially and therefore is not warranted. However, it would be useful to re-do this analysis with $\omega(t)$ that has a thicker tail since the probability mass of this function as described in figure 6 seems to drop off precipitously and therefore it seems that this distribution may be driving the result of section C.

We thank the Referee for this suggestion. We have performed a new analysis using a different possible definition of the curve ω (new SI Section A.3), which has a thicker tail. No significant changes in the model predictions have been observed.

7. What are the shaded areas in Figure 5 and are these just mean simulations?

The shaded areas represent the standard deviation. We have made it clear in the revised version of the manuscript.

References

- [RF1] L. Ferretti, C. Wymant, M. Kendall, L. Zhao, A. Nurtay, L. Abeler-Dörner, M. Parker, D. Bonsall, and C. Fraser. Quantifying SARS-CoV-2 transmission suggests epidemic control with digital contact tracing. *Science*, 2020.
- [RF2] C. Fraser, S. Riley, R. M. Anderson, and N. M. Ferguson. Factors that make an infectious disease outbreak controllable. *Proceedings of the National Academy of Sciences*, 101(16):6146–6151, 2004.
- [RF3] J. D. Murray. *Mathematical biology: I. An introduction*, volume 17. Springer, 2001.

Reviewers' Comments:

Reviewer #1:

Remarks to the Author:

I want to thank the authors for addressing my comments to the manuscript and for taking their time to improve the paper. I believe the current version is better than the previous, and I appreciate that the authors are more critical in their discussion of the efficacy of digital contact tracing apps. However, with the new version of the manuscript, a few issues have arisen, which the authors need to address before I can recommend it for publication.

Comments:

- The authors are more critical in their analysis of the results but there are still some shortcomings. At the end of section 3.1 they state that "these different scenarios highlight the efficacy of DCT", however, this is only under very specific conditions, which they do not explicitly mention. For instance, their results in SI section C.2, show that even one extra day of delay in recognizing a person is infected renders all contact tracing policies futile (according to results in the SI). They mention multiple places in the manuscript that results for $t = 2$ and 3 days greatly differ, but never explicitly mention the effect. This is a fundamental shortcoming which should be discussed directly in the manuscript and not be hidden away in the SI, since the results illustrate that digital contact tracing apps should only be implemented if all other containment factors are in place.

- The authors have greatly improved the quality of the manuscript by adding a section on how infectiousness depends on duration and proximity of contacts (section 2.2), however, the authors need to address an additional caveat here (and possibly in the discussion section as well). Infectiousness depends on the environment. For instance, a distance of 1.5 meters might be high-risk in an enclosed setting, while if the individuals are outside the risk will be much lower. According to the World Health Organization [1] places where COVID-19 spreads more easily are: i) crowded spaces with many people nearby, ii) close-contact settings (especially where people have close-range conversations), and iii) confined and enclosed spaces with poor ventilation. As such the authors' model ignores the effects of environmental conditions. An explicit discussion of this in the manuscript would create less confusion with policy makers.

- In the supplementary materials the authors have added a new section called "C.6 Compliance to quarantine decreases if notified multiple times". This section has a figure, Fig S12, which uses an extended version of the Copenhagen Network Study (2 months, instead of 1). Here the dynamics of Fig S12c have me worried. It seems the number of false negatives sharply drops. Is this because of the tracing strategies or because of the data? For instance, is there a vacation/exam period in the last 20 days of month 2?

I think the paper could benefit from including more details about the Copenhagen Network Study data. For instance, which month is the data for. This will also improve the reproducibility of their work.

Minor comments:

- In the abstract they write: "We find that restrictive policies are more effective in containing the *epidemics* but come at the cost of quarantining a large part of the population.", but I'm not sure if you want to use the plural version of the word "epidemic" here.

- I appreciate that the authors added additional details regarding their modeling framework and that they have added Equation 1. However, to improve readability, especially for readers not acquainted with math notation, the authors should explicitly explain what ρ means in the integral.

- Page 6 last sentence & page 7 first sentence. To improve readability the authors should consider

adding "according to some contact tracing strategy" such that the sentence becomes "... the fraction of the corresponding secondary cases that are actually quarantined *according to some contact tracing strategy*, ..."

- Page 7, sentence starting with "Each infected individual has ..." the *its* in "and its recent stored contacts ..." should be *their* instead.

- Page 10, last paragraph, sentence starting with "We then compare the set ..." they say "... performances of each tracing policies ... ", that is a strange wording. Do they mean "performance of tracing policies" or "performances of each tracing policy"?

- Page 12, second paragraph, first sentence, they write "... line with privacy preserving contact tracing app ...". Are they referring to a specific app or talking about apps in general?

- Page 14, second paragraph of section 3.2 is convoluted and should be rephrased.

- Figure 5, the column names in the table are confusing. Do they mean "percentage of population" and "percentage infected"?

- I took a quick glance at the list of references and found Ref 4 to be strangely formatted. Other references are also missing some information.

References

[1] <https://www.who.int/brunei/news/infographics---english>

Reviewer #2:
Remarks to the Author:

Thanks to the authors for a detailed response to my previous comments. The revised manuscript has significantly improved. I'd like to recommend the manuscript for publication once the additional comments below are addressed.

- 1) The current abstract of the manuscript is too generic. I recommend the authors to emphasize more on summarizing the major findings of the study and not waste too many words on laying the background and motivations of the study.
- 2) Supplementary Material, equation (S.1): It's not particularly clear to me why two scaling factors r_{R_0} and P_{R_0} are introduced. Is P_{R_0} used to calibrate R_0 in the absence of other NPIs (mask wearing, remote working etc) while r_{R_0} is used to represent the impact of those NPIs? The authors should clarify on this. If so, it's also not clear to me, for a given $w_{\text{exposure}}(e)$ and $w_{\text{dist}}(s_s)$ and a target $R_0=3$, how the parameter of P_{R_0} is inferred. It seems like the inference procedure is simulation based as it relies on the contact patterns of CNS. Did the authors adopt Latin-square sampling or it's a likelihood-based approach (i.e. particle filters/pMCMC)? However, this is not clearly presented in the manuscript SI, please clarify.
- 3) Supplementary Material, page 4, 5: I don't quite understand why the parameters of $w_{\text{exposure}}(e)$ and $w_{\text{dist}}(s_s)$ are correlated. For any given $w_{\text{exposure}}(e)$ and $w_{\text{dist}}(s_s)$ kernel, one could always adjust P_{R_0} to match the desired R_0 , am I correct? Maybe I'm missing something, and I hope the authors could clarify on this point.
- 4) Page 4, "First, it releases the assumption on the exponential behavior of infection growth or decline, allowing us to consider a generic form of the infection evolution..." and page 17 "(i) a general mathematical model that allows to evaluate the evolution of an epidemic in the presence of isolation and tracing for any shape of the epidemic growth (which can therefore be obtained numerically or by theoretical assumptions)" -> I don't agree that the authors could claim this novelty over the work by Fraser et al. Since in Fraser et al's work a generic framework without the assumption on growth was already laid out, exponential growth is just a asymptomatic behavior of the system... I recommend the authors to remove these claims.
- 5) Figure 1c, caption, "in all settings..", the author need to clarify which setting (the setting in Figure 1a or Figure 1b?).

Response to the Referees' reports

NCOMMS-20-27706

"Digital Proximity Tracing in the COVID-19 Pandemic on Empirical Contact Networks: Controlling re-emerging outbreaks"

Dear Editor,

Thank you for the opportunity to submit a revised manuscript. We have carefully gone through the reviewers' comments and found them to be helpful and insightful. Through responding to their feedback and criticisms, our manuscript has greatly improved. This holds true for the most recent round of suggested changes. The manuscript is now clearer and thus more useful to inform policy design. We sincerely thank the reviewers for their contribution to improving the work.

Below, we go over the comments from the two reviewers (*italic text*) point-by-point and provide a response to each specific comment, including a description of the changes implemented. We have highlighted modifications to the main text in blue. We have also slightly modified the title to improve its readability. The new title now reads as follows: "Digital proximity tracing on empirical contact networks: Controlling re-emerging outbreaks in the COVID-19 pandemic".

We hope that the reviewers agree that this new version of the manuscript is substantially improved and that the manuscript meets the requirements for publication in *Nature Communications*.

With many thanks and best regards,

G. Cencetti, G. Santin, A. Longa, E. Pigani, A. Barrat, C. Cattuto, S. Lehmann, M. Salathé,
B. Lepri

Contents

Major changes	2
Response to the report of Referee 1	3
Response to the report of Referee 2	6

Major changes

- We have modified the abstract in order to include more details on the results of the paper.
- We have modified figure S.12 in SI.
- We have written a new subsection A4 ("Contact pattern in the CNS data set") to describe the temporal evolution of the number of contacts in the CNS dataset, as well as a description of the extended time period used in Section C.6.
- We have changed several sentences in both main text and SI, as suggested by the referees, in order to improve their structure or readability. These sentences appear in blue text for easy tracking of the changes implemented.
- We have slightly modified the title to improve its readability. The new title now reads as follows: "Digital proximity tracing on empirical contact networks: Controlling re-emerging outbreaks in the COVID-19 pandemic".

Response to the report of Referee 1

I want to thank the authors for addressing my comments to the manuscript and for taking their time to improve the paper. I believe the current version is better than the previous, and I appreciate that the authors are more critical in their discussion of the efficacy of digital contact tracing apps. However, with the new version of the manuscript, a few issues have arisen, which the authors need to address before I can recommend it for publication.

We thank the Referee for the positive assessment of the content and contributions of our manuscript, and for the overall appreciation of its solidity and clarity.

The authors are more critical in their analysis of the results but there are still some shortcomings. At the end of section 3.1 they state that “these different scenarios highlight the efficacy of DCT”, however, this is only under very specific conditions, which they do not explicitly mention. For instance, their results in SI section C.2, show that even one extra day of delay in recognizing a person is infected renders all contact tracing policies futile (according to results in the SI). They mention multiple places in the manuscript that results for $t = 2$ and 3 days greatly differ, but never explicitly mention the effect. This is a fundamental shortcoming which should be discussed directly in the manuscript and not be hidden away in the SI, since the results illustrate that digital contact tracing apps should only be implemented if all other containment factors are in place.

We agree with the Referee and we have modified the sentence at the end of Section 3.1 highlighting that Digital Contact Tracing (DCT) only works in some constrained conditions. The result about the delay is now presented both in the abstract and in the Discussion section: in a modified sentence of Section 4.1 (at page 17) and in Section 4.2, where it originally appeared.

The authors have greatly improved the quality of the manuscript by adding a section on how infectiousness depends on duration and proximity of contacts (section 2.2), however, the authors need to address an additional caveat here (and possibly in the discussion section as well). Infectiousness depends on the environment. For instance, a distance of 1.5 meters might be high-risk in an enclosed setting, while if the individuals are outside the risk will be much lower. According to the World Health Organization (<https://www.who.int/brunei/news/infographics---english>) places where COVID-19 spreads more easily are: i) crowded spaces with many people nearby, ii) close-contact settings (especially where people have close-range conversations), and iii) confined and enclosed spaces with poor ventilation. As such the authors’ model ignores the effects of environmental conditions. An explicit discussion of this in the manuscript would create less confusion with policy makers.

We thank the Referee for this suggestion. It is correct that our model does not provide for a differentiation between indoor and outdoor transmission, and it would certainly be interesting to take into account this factor in a model for contagion. Two important points however hinder this model refinement. First, no dataset describing contacts between individuals at the level of detail that we consider here is currently available, at least for sizes and durations comparable to the CNS dataset. Second, actual digital contact tracing is based on mobile phone apps that usually do not have access to the environment characteristics. Implementation of policies concerning the definition of at-risk contacts can therefore not take this factor into account. For these reasons, we have focused our work only on duration and proximity of contacts. We have nonetheless added a sentence in the text at the beginning of page 20 to address and discuss this point.

In the supplementary materials the authors have added a new section called “C.6 Compliance to quarantine decreases if notified multiple times”. This section has a figure, Fig S12, which uses an

extended version of the Copenhagen Network Study (2 months, instead of 1). Here the dynamics of Fig S12c have me worried. It seems the number of false negatives sharply drops. Is this because of the tracing strategies or because of the data? For instance, is there a vacation/exam period in the last 20 days of month 2? I think the paper could benefit from including more details about the Copenhagen Network Study data. For instance, which month is the data for. This will also improve the reproducibility of their work.

We thank the Referee for pointing out this fact. Indeed, the sharp drop is due to a corresponding drop in the number of contacts (see Figure 1) that is probably due to a holiday week. We have performed new simulations by removing the week with no contacts and extending the dataset by an additional week, in order to avoid this week with very few contacts. In this way, the whole timespan used for the simulations has an amount of contacts that remains on average homogeneous in time. We replaced Figure S.12 in SI accordingly.

Additionally, a new subsection A4 in the SI describes the dataset in more details.

Figure 1: Temporal evolution of the total degree (i.e., number of contacts) of the static networks in the two months of the CNS dataset used in Section C.6 in the previous submission. A sharp drop of the number of contacts can be observed in the penultimate week.

Minor comments:

1. In the abstract they write: “We find that restrictive policies are more effective in containing the epidemics but come at the cost of quarantining a large part of the population.”, but I’m not sure if you want to use the plural version of the word “epidemic” here.

We have modified the abstract according to the Referee’s suggestion.

2. I appreciate that the authors added additional details regarding their modeling framework and that they have added Equation 1. However, to improve readability, especially for readers not acquainted with math notation, the authors should explicitly explain what ρ means in the integral.

We have added a sentence just below the equation, in order to improve readability: “where the integration variable ρ spans the time range between 0 and $t - \tau$, meaning that the contagion at time t from people infected at time $t - \tau$ is in turn affected by contagion at time ρ before $t - \tau$.”

3. Page 6 last sentence and page 7 first sentence. To improve readability the authors should consider adding "according to some contact tracing strategy" such that the sentence becomes "... the fraction of the corresponding secondary cases that are actually quarantined **according to some contact tracing strategy, ...**"

We have modified this sentence according to the Referee's suggestion.

4. Page 7, sentence starting with "Each infected individual has ..." the **its** in "and its recent stored contacts ..." should be **their** instead.

We have fixed this issue in the revised version of the manuscript by rephrasing the sentence.

5. Page 10, last paragraph, sentence starting with "We then compare the set ..." they say "... performances of each tracing policies ...", that is a strange wording. Do they mean "performance of tracing policies" or "performances of each tracing policy"?

We have fixed this issue in the revised version of the manuscript.

6. Page 12, second paragraph, first sentence, they write "... line with privacy preserving contact tracing app ...". Are they referring to a specific app or talking about apps in general?

We were referring to apps in general. We have rephrased the sentence.

7. Page 14, second paragraph of section 3.2 is convoluted and should be rephrased.

We have rephrased the sentence.

8. Figure 5, the column names in the table are confusing. Do they mean "percentage of population" and "percentage infected"?

We have clarified the wording in the revised version of the manuscript.

9. I took a quick glance at the list of references and found Ref 4 to be strangely formatted. Other references are also missing some information.

Good point, we have fixed Ref 4 and remaining references in the revised version of the manuscript.

Response to the report of Referee 2

Thanks to the authors for a detailed response to my previous comments. The revised manuscript has significantly improved. I'd like to recommend the manuscript for publication once the additional comments below are addressed.

We thank the Referee for the careful reading and for appreciating our contribution to this area of research. We have revised the manuscript to improve it according to the Referee's suggestions.

The current abstract of the manuscript is too generic. I recommend the authors to emphasize more on summarizing the major findings of the study and not waste too many words on laying the background and motivations of the study.

We have modified the abstract, including details regarding the new findings of our study.

Supplementary Material, equation (S.1): It's not particularly clear to me why two scaling factors rR_0 and PR_0 are introduced. Is PR_0 used to calibrate R_0 in the absence of other NPIs (mask wearing, remote working etc) while rR_0 is used to represent the impact of those NPIs? The authors should clarify on this. If so, it's also not clear to me, for a given $\omega_{exposure}(e)$ and $\omega_{dist}(ss)$ and a target $R_0 = 3$, how the parameter of PR_0 is inferred. It seems like the inference procedure is simulation based as it relies on the contact patterns of CNS. Did the authors adopt Latin-square sampling or it's a likelihood-based approach (i.e. particle filters/pMCMC)? However, this is not clearly presented in the manuscript SI, please clarify.

The two points raised by the Referee can be explained as follows:

- The use of two different parameters is mainly for modelling convenience. Indeed, as correctly pointed out by the Referee, one parameter would be sufficient and the only quantity that matters is the product of p_{R_0} and r_{R_0} . However, we decided to adopt a two stage approach: First, given the curves $\omega_{distance}$ and $\omega_{exposure}$, we calibrate p_{R_0} so that the value of the empirical R_0^{data} coincides with the target value $R_0 = 3$. In a second stage we calibrate the second parameter r_{R_0} to reduce R_0^{data} to the desired values of 1.2, 1.5, 2. This is of course exactly equivalent to a simpler approach where a single parameter is used, but our strategy makes it more straightforward to explore the parameter space to find suitable values. This is explained at the end of Section A.2 in SI.
- To infer the value of R_0^{data} given all the other parameters, we use the approach of [RF1], i.e., "we computed the value of R_0 as the mean, over different realizations, of the number of secondary cases from the single initial randomly chosen infectious individual" (see [RF1]). In this paper we use a repetition over 800 random initializations for each value of R_0 . As an example, we report in Figure 2 the complete distribution obtained in the estimation of $R_0 = 3$. This is clarified in the new version of the manuscript at the beginning of Section A.2 in SI.

Supplementary Material, page 4, 5: I don't quite understand why the parameters of $\omega_{exposure}(e)$ and $\omega_{dist}(ss)$ are correlated. For any given $\omega_{exposure}(e)$ and $\omega_{dist}(ss)$ kernel, one could always adjust PR_0 to match the desired R_0 , am I correct? Maybe I'm missing something, and I hope the authors could clarify on this point.

Good point. It is correct: $\omega_{exposure}$ and ω_{dist} are defined as two independent functions, respectively reflecting the dependency from duration and proximity. We chose, however, to set their

Figure 2: The histogram reports the values of R_0 obtained by simulating the contagion on the network of contacts CNS without any isolation and quarantine for $p_{R_0} = 60$ and $r_{R_0} = 1$. With 800 simulations we observe that almost half of the times we have zero contagions and a second peak appears around 3, with a mean value of $\langle R_0 \rangle = 3.11$ (vertical line).

free parameters combining the two effects/aspects of infectiousness. The reason for this choice was to explore how their mutual contribution change in shaping the contagions. Therefore we kept p_{R_0} fixed and varied duration and proximity parameters, observing different scenarios where the responsibility of contagions is tuned between longer durations and closer proximities. In the revised SI, we have added a comment to clarify this point (at the end of section A.2).

Page 4, "First, it releases the assumption on the exponential behavior of infection growth or decline, allowing us to consider a generic form of the infection evolution..." and page 17 "(i) a general mathematical model that allows to evaluate the evolution of an epidemic in the presence of isolation and tracing for any shape of the epidemic growth (which can therefore be obtained numerically or by theoretical assumptions)" → I don't agree that the authors could claim this novelty over the work by Fraser et al. Since in Fraser et al's work a generic framework without the assumption on growth was already laid out, exponential growth is just a asymptomatic behavior of the system... I recommend the authors to remove these claims.

We have changed both sentences as suggested.

Figure 1c, caption, "in all settings..", the author need to clarify which setting (the setting in Figure 1a or Figure 1b?).

We modified the sentence by clarifying that those settings are the three configurations presented in the same figure.

References

- [RF1] J. Stehlé, N. Voirin, A. Barrat, C. Cattuto, V. Colizza, L. Isella, C. Régis, J.-F. Pinton, N. Khanafer, W. Van den Broeck, et al. Simulation of an seir infectious disease model on the dynamic contact network of conference attendees. *BMC medicine*, 9(1):87, 2011.

Reviewers' Comments:

Reviewer #1:

Remarks to the Author:

I wish to thank the authors for their detailed response to my comments. I am glad to see the authors more critically discuss DCT and its efficacy for containing the COVID-19 pandemic. The revised manuscript has significantly improved. I recommend it for publication.

Reviewer #2:

Remarks to the Author:

The authors responses has addressed my previous concerns. I do not have further comments.